# Carbonylolysis of waste polyesters into high-value organic acids

Dongxu Liu[1,2], Siming Zhu[1,2] & Qingqing Mei [1,2]✉

Polyesters such as PET contribute substantially to global plastic waste, yet current recycling approaches are hindered by high energy demands, inefficient product separation, and limited valorization pathways. We report a one-pot "carbonylolysis" strategy that couples polyester depolymerization with in situ carbon-chain reconstruction, producing high-value $C_3^+$ carboxylic acids under relatively mild conditions (170 °C, 2 MPa CO). Using a Rh–iodide catalyst, PET is quantitatively converted to terephthalic acid (99%) and propionic acid (96%). Mechanistic studies show that ethylene glycol released from PET hydrolysis undergoes iodide-assisted elimination followed by Rh-catalyzed carbonylation. The method applies broadly to diverse polyester wastes, including textiles and bio-based plastics. Life-cycle assessment and techno-economic analysis reveal substantial gains in energy efficiency, carbon footprint reduction, and wastewater minimization over conventional recycling routes. By integrating molecular-level reconstruction into polyester recycling, carbonylolysis establishes a sustainable blueprint for converting waste polyesters into high-value carboxylic acid.

Plastics are essential to contemporary society and industrial processes, with widespread use across packaging, electronics, and construction due to their durability and malleability[1,2]. However, the majority of plastic waste is either incinerated (19%) or landfilled (49%), raising pressing concerns regarding resource loss, environmental pollution, and greenhouse gas emissions[3,4]. Polyethylene terephthalate (PET) is the most widely used polyester and a major contributor to plastic waste due to its extensive use and slow degradation[5,6]. Mechanical recycling remains the dominant strategy for managing PET waste[7,8], but suffers from material downgrading and limited feedstock tolerance. Chemical recycling methods such as pyrolysis, solvolysis and hydrogenolysis offer alternatives for recovering monomers or converting waste into value-added chemicals[9–11]. Among them, hydrolysis stands out as the most direct and atom-efficient route to break PET into its constituent monomers, terephthalic acid (TPA) and ethylene glycol (EG)[12,13] (Fig. 1a–l). However, its industrial adoption is constrained by harsh reaction conditions, high energy consumption, and laborious downstream separation[14,15].

These limitations have prompted increasing interest in one-step catalytic processes that couple depolymerization with the simultaneous conversion of hydrolysis intermediates into high-value products. Especially, EG derived from PET hydrolysis presents a key opportunity for value enhancement within a circular economy framework. Direct EG capture and conversion afford a wide range of chemicals containing the $-OCH_2CH_2O-$ motif[9,16], but they are usually constrained by the reaction equilibria. More extensively studied approaches circumvent this limitation by exploiting the intrinsic reductive nature of EG, which enables dehydrogenation with concomitant $H_2$ release or transfer hydrogenation[17,18], typically co-producing oxygenates or carbon. Selective C–C bond cleavage of EG affords $C_1$ oxygenates such as formic acid[19,20], while backbone-preserving pathways yield $C_2$ products including glycolic and acetic acids[21–23]. However, these PET-derived EG conversion routes generally rely on excess strong base, as EG dehydration is typically promoted under strongly basic conditions across thermal, electrochemical, and photochemical methods. Moreover, nonthermal strategies usually

[1]State Key Laboratory of Soil Pollution Control and Safety, Zhejiang University, Hangzhou, China. [2]Institute of Environment Science and Technology, College of Environmental and Resource Sciences, Zhejiang University, Hangzhou, China. ✉e-mail: meiqq@zju.edu.cn

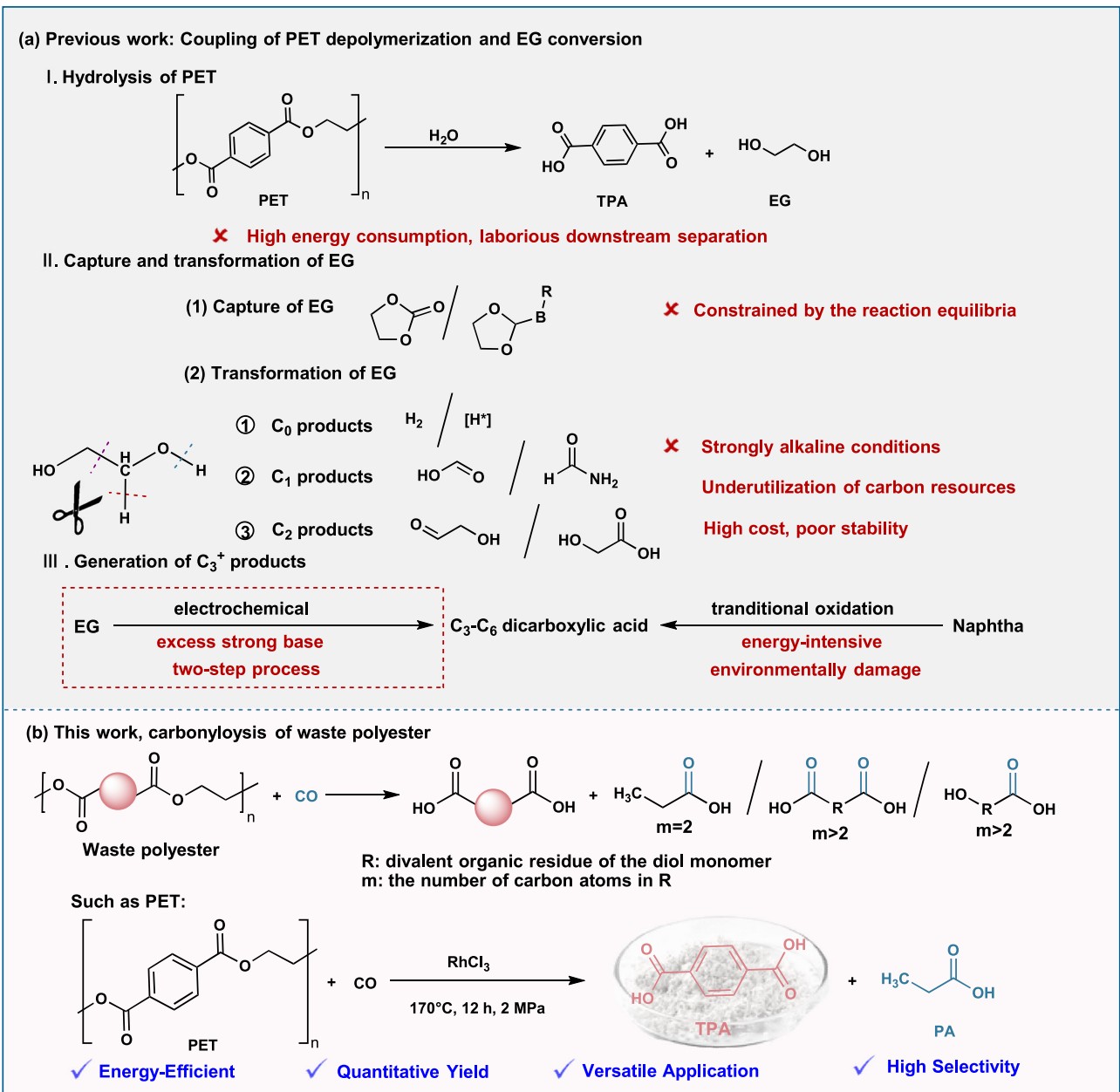

**Fig. 1 | Schematic overview of waste polyester depolymerization. a** Hydrolysis of PET and subsequent transformation of the liberated EG. **b** Proposed carbonylolysis strategy for valorization of waste polyesters, highlighting PET as a representative substrate.

require prior PET depolymerization into EG, which also involves large base inputs[19,24]. Notably, excess base typically converts TPA and other acidic products into salts that require re-acidification, consuming large amounts of acid and base while generating substantial saline waste-water, complicating integration into practical recycling systems (Fig. 1a-II). Therefore, overcoming this reliance on excess base remains a core challenge for sustainable PET upcycling.

To date, the product spectrum from PET-derived EG has been largely confined to low-carbon ($C_0$–$C_2$) molecules[11,19–23]. Crucially, C–C chain-extended $C_3^+$ products are rarely pursued[11,25,26], despite their markedly higher industrial value. $C_3$–$C_6$ carboxylic acids, such as propionic, glutaric, and adipic acids, remain predominantly fossil-derived via energy- and emissions-intensive oxidation processes[27–29], representing especially appealing targets for sustainable production (Fig. 1a-III). This gap underscores the urgent need for carbon-reconstructive catalytic pathways that can upgrade waste-derived diols into longer-chain carboxylic acids. Such pathways must combine

selective bond scission, controlled C–C coupling, and oxygenate functionalization, ideally under low- or no-base conditions. Achieving this would unlock higher-value products and significantly broaden the chemical space accessible from plastic waste within sustainable recycling systems.

Herein, we report a previously unexplored carbonylation-assisted depolymerization strategy, termed "carbonylolysis", that couples PET backbone cleavage with in situ carbon chain reconstruction of EG (Fig. 1b). Carbonylation offers a highly atom-economy, regioselective, and thermodynamically favorable route for upgrading alcohols into longer-chain acids[30], with its highly exothermic nature supplying thermal energy to drive self-sustaining processes. Crucially, as a fundamental $C_1$ building block, carbon monoxide (CO) can be sustainably sourced through biomass conversion, waste valorization, or $CO_2$ reduction[31,32]. By integrating carbonylation with polyester depolymerization, we establish a single-step process that converts PET into both TPA and high-value $C_3^+$ acids under mild, scalable conditions.

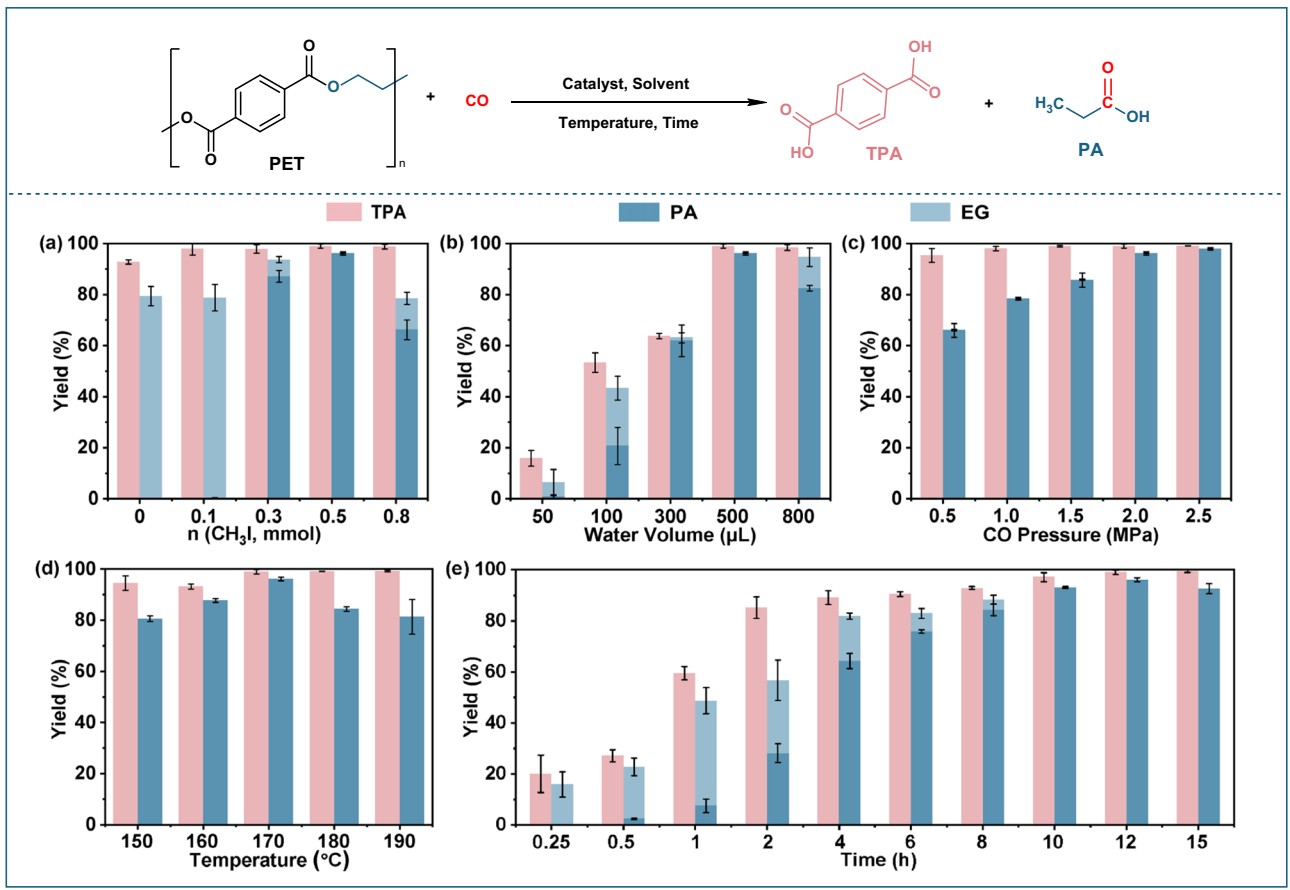

**Fig. 2 | Construction and optimization of the catalytic system.** Reaction conditions: PET (0.192 g, 1 mmol structural unit), HFIP (2 mL), $H_2O$ (0.5 mL), $CH_3I$ (0.5 mmol), CO (2 MPa), and $RhCl_3$ (5 mol%) at 170 °C for 12 h. Effects of reaction parameters on performance: (**a**) $CH_3I$ amount; (**b**) water volume; (**c**) CO pressure; (**d**) temperature; (**e**) reaction time. Error bars represent the standard deviations from the means of at least three repeated experiments.

Typically, at 170 °C and 2 MPa CO over 12 h, PET carbonylolysis delivers 99% TPA recovery and 96% conversion of EG into propionic acid. The process avoids terephthalate salt formation, minimizes acid/base usage, and drastically reduces wastewater generation. Furthermore, the strategy is broadly applicable to various polyester wastes, affording glutaric acid, adipic acid, and related compounds in high yields. By shifting the paradigm from fragmentative degradation to skeletal reconstruction, carbonylolysis offers a blueprint for circular polymer valorization, expanding product diversity, improving carbon efficiency, and delivering a sustainable route for valorizing waste plastic into valuable carboxylic acids.

## Results

### Catalytic system development for PET carbonylolysis

The proposed carbonylolysis process integrates simultaneous PET depolymerization with carbonyl insertion into the hydrolysis-derived diol fragments. To identify an effective catalytic system, we systematically screened a range of metal catalysts ($RhCl_3$, $RuCl_3$, $IrCl_3$, $PdCl_2$, $CoCl_2$, $NiCl_2$), iodide additives (LiI, NaI, $ZnI_2$, $CH_3I$, and CuI), and solvents commonly employed in carbonylation chemistry, including toluene (TL), water, acetonitrile (ACN), hexafluoroisopropanol (HFIP), cyclohexane (CYH), acetic acid (HAc)[33], as well as the commonly used bio-based solvents dimethyl isosorbide (DMI) and γ-valerolactone (GVL). Among the tested metals, $Rh^{3+}$ exhibited the highest catalytic activity for PET carbonylolysis (Supplementary Fig. S2), while the effect of the chloride counter-anion was negligible (Supplementary Table S1). Both iodide source and solvent were identified as critical parameters governing reaction efficiency[34,35]. $CH_3I$ and HI is the most effective

iodide additives, owing to their ability to activate the Rh catalyst and sustain a high free-iodide concentration under reaction conditions (Supplementary Fig. S3). HFIP was identified as the optimal solvent (Supplementary Fig. S4), benefiting from its excellent PET-dissolving capability and high compatibility with the carbonylolysis system[36,37].

In this transformation, terephthalic acid (TPA) is the primary product from PET depolymerization, while propionic acid (PA) is formed via carbonyl insertion into EG, the diol released from PET. Using commercially sourced, low-shear–pulverized PET powder as the feedstock (without additional ball-milling pretreatment that could alter the polymer structure[38]), we systematically optimized key reaction parameters to maximize PA yield, including the amounts of $CH_3I$, $H_2O$, and $Rh^{3+}$, as well as the CO pressure (Fig. 2a–c, Supplementary Fig. S5).

The presence of an iodide source was essential for carbonylation activity, as a sufficiently high iodide concentration is required to stabilize the catalytically active Rh–iodide carbonyl species[39–42]. At low $CH_3I$ loadings, iodide activity was insufficient to sustain the active Rh speciation, resulting in incomplete carbonylation and predominant formation of EG (Fig. 2a). As $CH_3I$ is increased into an intermediate regime, the steady-state population of catalytically competent anionic Rh–iodide–carbonyl species is established, leading to a sharp rise in PA formation (Supplementary Table S2). Further increases in $CH_3I$ result in a plateau in activity and selectivity, indicative of saturation of the active Rh species. At much higher $CH_3I$ loadings, excess iodide overcompetes with CO for coordination, driving Rh speciation toward iodide-saturated CO-deficient environments and limiting CO insertion, which reduces PA yield (Fig. 2a)[43]. This behavior is fully consistent with

established Rh−iodide carbonylation chemistry[34,44]. Water addition was also found to be crucial (Fig. 2b). Small amounts of $H_2O$ significantly promoted PET depolymerization and subsequent carbonylation. Increasing $H_2O$ content enhanced EG and PA formation initially (EG increased from 0.3% to 18%, PA from 1.5% to 28%). Further addition of water led to near-complete consumption of EG and a substantial increase in PA yield reaching 96%. However, excessive water compromised the reaction by hydrolyzing anionic Rh(I) carbonyl species into less stable species (e.g., $[Rh(CO)_2I_2]^-$ to $[Rh(CO)(H_2O)I_2]^-$ [45]), and promoting the water-gas shift reaction, reducing CO availability and decreasing the PA yield to 81%. CO pressure was another key parameter. Increasing the CO pressure from 0.5 MPa to 2.0 MPa progressively improved the PA yield from 69% to 96% (Fig. 2c), though little improvement was observed beyond 2.0 MPa. Therefore, 2.0 MPa was chosen as the optimal pressure of CO. Likewise, PA yield increased with $Rh^{3+}$ content up to 0.05 mmol, and further increases provided minimal additional benefit (Supplementary Fig.S5). Reaction temperature and time were then optimized (Fig. 2d, e). Increasing the temperature from 150 °C to 170 °C significantly improved PA yield from 80% to 96% (Fig. 2d). However, due to the highly exothermic nature of the carbonylation process, further temperature increases reduced the PA yield to 76%. Kinetic profiling under optimal conditions (170 °C, 2.0 MPa CO, 0.05 mmol $Rh^{3+}$, 0.5 mmol $CH_3I$, 500 μL $H_2O$) showed a progressive increase in TPA yield, which stabilized above 99% over time (Fig. 2e). EG concentration initially increased and then declined, while PA yield gradually rose from 2.2% to 96% confirming effective coupling of EG with CO. These results demonstrate the viability of a $Rh^{3+}$/I/HFIP system in mediating tandem PET depolymerization and EG carbonylation, achieving near-quantitative yields of both TPA and PA.

## Substrate scope and real-world waste plastic compatibility

The robustness of the carbonylolysis strategy was demonstrated using a range of commercial PET samples containing common additives and contaminants. Six types of real-world PET waste (including beverage bottles, non-woven fabrics, colored trays, transparent films, garments and woven materials) were successfully upcycled, affording PA yields of 90–98% and TPA yields of 92–98% (Fig. 3a). The recycling of blended textiles presents significant challenges due to the frequent incorporation of cotton, spandex, nylon, and synthetic dyes. Remarkably, even PET fabrics blended with cotton (15–35%), spandex (5%), chinlon (35%), or rayon (43%) achieved high product yields (PA: 89–93%; TPA: 90–94%) under the optimized conditions (Fig. 3b). Notably, all PET samples were manually cut into small pieces without further powdering or micronization, minimizing potential structural modification associated with intensive mechanical pretreatments such as ball milling[38]. Further size-activity experiments demonstrate that PET powdering is unnecessary for efficient conversion under the present HFIP-mediated conditions (Supplementary Table S3 and Supplementary Fig. S7), owing to the strong solubility of HFIP toward PET, which enables the direct processing of real waste streams without energy-intensive size reduction. The scope of this strategy was further extended to various polyester substrates, including polyethylene 2,5-furandicarboxylate (PEF), polyethylene glycol succinate (PES), polyethylene glycol (PEG), polyethylene adipate (PEA), polytrimethylene terephthalate (PTT), polybutylene terephthalate (PBT) and polybutylene adipate-co-terephthalate (PBAT). The process proved highly efficient across this diverse range of glycol- and diol-derived polyesters. For glycol-based polyesters such as PEF, PES, PEG, and PEA, both the carbonylation product (PA) and the hydrolysis-derived monomer were obtained in yields exceeding 90%. Polyesters based on longer-chain diols (PTT, PBT, PBAT) also underwent efficient transformation. In these cases, where the hydroxyl-bearing carbon atoms are non-adjacent, the reaction yields dicarboxylic or hydroxy acids as products[41], alongside high TPA recovery, with both product classes exceeding 90% yield (Fig. 3c).

## Kinetic dissection of the hydrolysis−carbonylation cascade

Understanding the mechanism of PET carbonylolysis is essential to rationalize its high selectivity and efficiency. Conceptually, the carbonylation reaction may proceed via activation of either the ester functionalities in the PET backbone or the hydroxyl groups of the depolymerized intermediate, ethylene glycol (EG). To differentiate these possibilities, we systematically investigated the carbonylation reactivity of PET, bis(2-hydroxyethyl) terephthalate (BHET), and EG under identical reaction conditions. All three substrates yielded comparable amounts of PA (Fig. 4a−c and Supplementary Table S4), strongly suggesting that PET is first hydrolyzed to release EG, which is subsequently carbonylated. The absence of byproducts such as 3-hydroxypropionic acid (3-HP) or succinic acid (SA), which would otherwise result from direct ester carbonylation, further supports this sequential pathway. Based on these findings, the overall carbonylolysis process can be divided into two distinct yet coupled stages: (i) hydrolytic depolymerization of PET into EG and TPA, and (ii) catalytic carbonylation of EG into $C_3^+$ organic acids. To gain further insights, we performed time-resolved kinetic studies under optimized conditions (Fig. 4), which revealed important distinctions in the dynamics and rate-limiting features of the two stages.

Due to its inertness and sluggish hydrolysis kinetics, PET remained largely undecomposed in water at 170 °C, even in the presence of $Rh^{3+}$ alone (Fig. 4d-I). To enhance depolymerization, we employed HFIP, a hydrogen-bonding solvent known to disrupt polymer crystallinity and improve the solubility of esters and polyamides[46,47]. HFIP significantly promoted PET dissolution and hydrolysis, achieving a 43% yield of TPA after 4 hours at 170 °C (Fig. 4d-I). The addition of a small amount (50 μL) of PA, a downstream product of EG carbonylation, further elevated the TPA yield to 48%, likely due to acid-promoted ester hydrolysis[48]. Under optimized carbonylolysis conditions with CO, the TPA yield increased to 86%, and kinetic analysis showed a significantly accelerated hydrolysis rate compared to conventional systems (Fig. 4d-I). This improvement arises from the coupling of EG carbonylation, which not only upgrades a hydrolysis product but also thermodynamically pulls the depolymerization equilibrium forward (Supplementary Table S7). These results underscore the dual importance of HFIP-mediated solubilization and product-driven equilibrium shifting in Stage 1.

Turning to Stage 2, kinetic profiles revealed that the concentrations of both EG and TPA increased rapidly at the early stage of reaction, with TPA quickly plateauing, indicative of fast and near-complete hydrolysis (Fig. 4d-II). In contrast, EG concentration exhibited a transient peak followed by gradual decline, while PA concentration steadily increased throughout the reaction. These profiles strongly suggest that EG carbonylation is the rate-limiting step in the overall process. Interestingly, the measured TPA yield temporarily exceeded the combined yields of EG and PA, implying that a fraction of EG followed an alternative transformation route beyond direct carbonylation. Supporting this, NMR analysis detected the formation of 1,2-diiodoethane within the first hour of the reaction (Supplementary Fig. S9), and GC analysis identified ethylene as a transient intermediate that appeared shortly after EG and was nearly completely converted to PA within 12 hours (Fig. 4d-II). These observations align with literature reports indicating that vicinal diols can undergo iodide-mediated substitution and β-elimination to generate alkenes[41,49]. The role of 1,2-diiodoethane as a key intermediate was further confirmed by independent reactions using it as a substrate, which also produced PA in high yield (Supplementary Fig. S10). Mechanistically, iodide species serves a dual function in carbonylation reaction: activating EG for substitution and stabilizing the Rh catalytic species[39]. When the iodine dosage was insufficient (e.g., 0.1 mmol, Fig. 2a), PA formation was significantly suppressed, highlighting its essential role in facilitating this reaction. Moreover, control experiments without $Rh^{3+}$ led to trace levels of ethylene and negligible PA formation (Fig. 4d-III). This

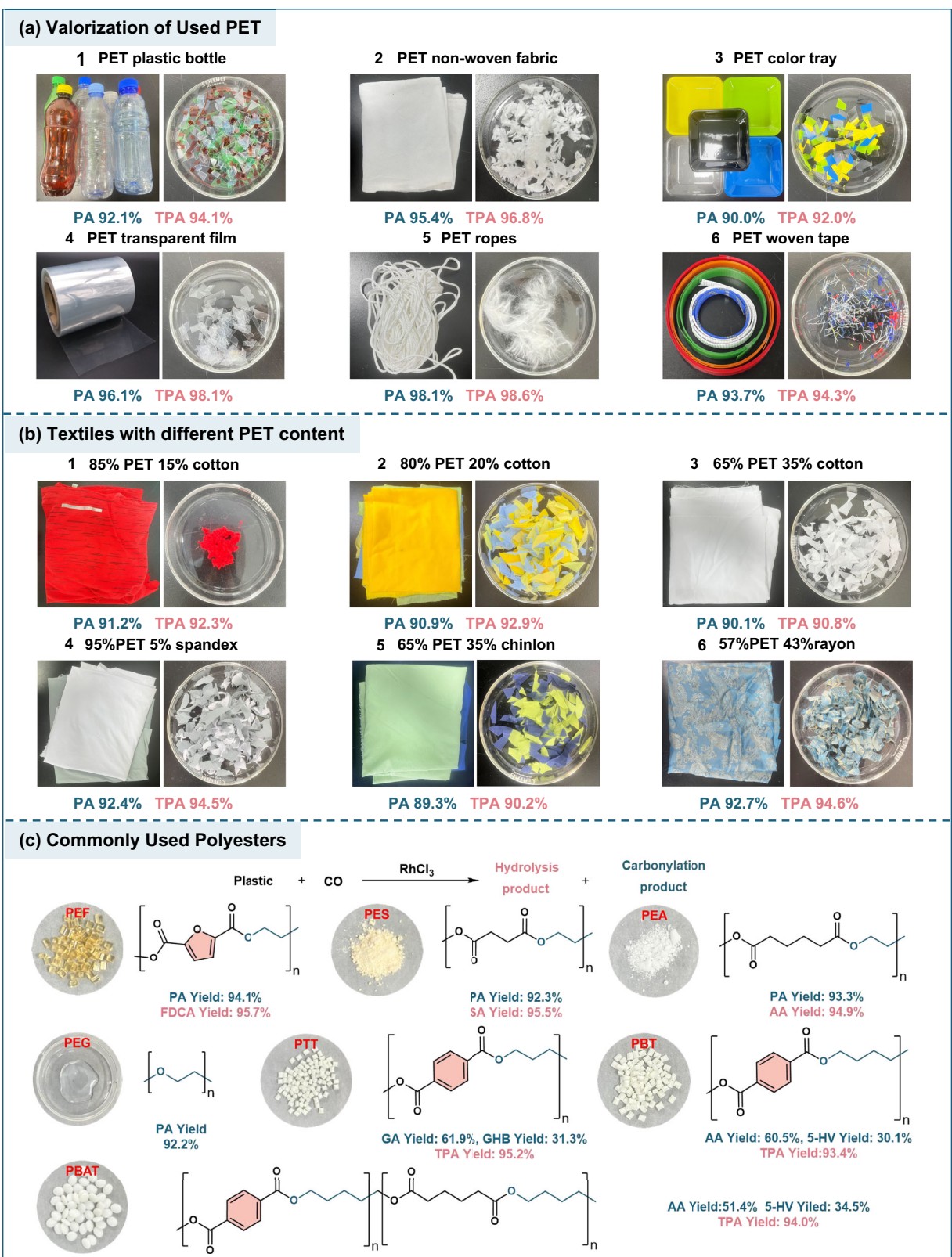

**Fig. 3 | Substrate scope and practical application of the carbonylolysis system.** Standard reaction conditions: PET (0.192 g, 1 mmol structural unit), HFIP (2 mL), $H_2O$ (0.5 mL), $CH_3I$ (0.5 mmol), CO (2 MPa), and $RhCl_3$ (5 mol%) at 170 °C for 12 h. Substrate evaluation: (**a**) real waste PET; (**b**) textiles with varying PET content; (**c**) different polyesters. Abbreviations: *FDCA* 2,5-furandicarboxylic acid; *SA* succinic acid; *AA* adipic acid; *GA* glutaric acid; *GHB* 4-hydroxybutyric acid; *5-HV*, 5-hydroxyvaleric acid.

confirms that both iodide and Rh catalysis play indispensable catalytic roles in the carbonylative transformation[50]. In addition, coupling with the carbonylation reaction facilitates ethylene generation to some extent. Collectively, these findings support a tandem cascade mechanism in which EG is first converted to 1,2-diiodoethane, followed by elimination to ethylene, and subsequently Rh-catalyzed carbonylation to yield PA. The reaction kinetics, particularly the delayed consumption of EG and gradual PA formation, further suggest that the iodide-mediated substitution/elimination sequence constitutes the rate-determining step in the overall process.

## Selectivity: β-elimination vs oxidative carbonylation

Notably, this mechanistic route presents a surprising ambiguity: one might anticipate that carbonylation of EG would yield 3-hydroxypropionic acid (3-HP) or succinic acid (SA) rather than PA. To clarify this, we performed density functional theory (DFT) calculations to map the competing reaction pathways and their associated Gibbs free energy barriers (Fig. 4e). The analysis focused on the rate-determining iodide-mediated substitution and elimination steps, starting from 1,2-diiodoethane (**IM1**), formed via substitution of EG's hydroxyl groups by iodide[49]. From **IM1**, two distinct reaction channels diverge: indirect carbonylation via ethylene formation through β-elimination[41] (Pathway A)[51], as suggested by kinetic analysis, and direct carbonylation via oxidative addition of the organic iodide (Pathway B)[40], in line with established halide carbonylation chemistry. The catalytically active $[HRh(CO)I_3]^-$ and $[Rh(CO)_2I_2]^-$ species coexist and are well documented to promote olefin and organic halide carbonylation, respectively[39,40]. In Pathway A, **IM1** undergoes iodide-assisted β-elimination of $I_2$ via **TS1** to generate ethylene (**IM2**). This ethylene then coordinates to the Rh center of $[HRh(CO)I_3]^{-[40]}$ to form **IM2'**, followed by protonation via **TS2** to yield the alkyl–Rh intermediate (**IM3**). Subsequent CO insertion through **TS3** produces the acyl species (**IM4**), which, upon hydrolysis, releases PA. In this pathway, β-elimination (**IM1** → **TS1**) is the rate-determining step, with a calculated free-energy barrier of 29.9 kcal/mol. By contrast, Pathway B involves direct oxidative addition of **IM1** to $[Rh(CO)_2I_2]^{-[39]}$. **IM1** first coordinates to the Rh center, after which oxidative insertion into the C−I bond occurs via **TS2'** to form the six-coordinate intermediate **IM3'**. Subsequent CO insertion through **TS3'** affords the acyl intermediate (**IM4'**), which, upon hydrolysis, generates 3-HP. This product can undergo further analogous transformations to yield SA. The rate-determining step in this pathway is oxidative addition (**IM1** → **TS2'**), associated with a substantially higher free-energy barrier of 40.8 kcal/mol. The markedly lower barrier of the β-elimination pathway (29.9 vs. 40.8 kcal/mol) explains the selective formation of PA while suppressing $C_3$ hydroxy acids and succinic acid. For polyesters lacking a β-diol motif, hydroxy acids or diacids dominate, consistent with the absence of an accessible β-elimination pathway (Fig. 3).

Integrating kinetic, experimental, and computational evidence, we propose the following overall mechanism for the carbonylolysis process. PET is initially hydrolyzed to TPA and EG. In the presence of iodide, EG is converted to 1,2-diiodoethane, which undergoes β-elimination to ethylene rather than direct carbonylation. This ethylene is subsequently carbonylated to form PA. This cascade reconstructs the carbon backbone via carbonyl insertion, achieving high product selectivity and excellent atom economy (Fig. 4).

## Economic and environmental assessment of carbonylolysis

Process design is crucial for evaluating reaction feasibility, separation strategies, and system scalability[52,53]. Given that the $CH_3I$–CO system generates acetic acid, which increases separation energy demands and leads to carbon loss[44], we evaluated HI as a more sustainable alternative. Remarkably, the addition of 0.4 mmol of HI afforded >99% yield of TPA and 93% yield of PA, confirming its high catalytic efficacy (Supplementary Table S8). In homogeneous carbonylation reactions,

the stability of Rh catalysts is strongly dependent on CO partial pressure. At low CO pressure, the active Rh species tend to form $RhI_3$ precipitates, resulting in catalyst deactivation and metal loss from the reaction system[54]. Based on this mechanistic understanding, we designed a separation protocol (Fig. 5a) in which solid TPA is first recovered by pressure filtration under CO, followed by atmospheric filtration of the remaining solution to recover the Rh catalyst with removal of CO. PA is subsequently isolated by vacuum distillation of the mother liquor. CO is stoichiometrically consumed during carbonylation, and excess CO can be readily recovered and recycled within the process. We conducted laboratory-scale cycling experiments to simulate and validate the PET carbonylolysis process (Supplementary Fig. S11), which served as the basis for further developing a preliminary process scheme (Supplementary Fig. S15). To assess the broader viability of this strategy, we performed a comprehensive system-level evaluation, including mass and energy balances (Supplementary Fig. S15 and Supplementary Tables S9–S12), life cycle assessment (LCA), and techno-economic assessment (TEA).

Full LCA results are presented in Supplementary Tables S13–S15, with the system boundary illustrated in Fig.5b. Specifically, CO was treated as a purchased industrial feedstock, and its upstream environmental impacts were fully incorporated in the LCA model[55,56]. Compared to conventional routes, the carbonylolysis process not only generates high-value products and circumvents the need for EG separation, but also substantially reduces non-renewable energy use (NREU) and greenhouse gas emissions (GWP). As shown in Fig. 5c, converting PET waste into TPA and PA dramatically lowers environmental impact relative to incineration or landfill (baseline: 10.51 kg $CO_{2-eq}$/kg PET)[57]. In particular, the process achieved NREU values of 22.9 MJ/kg PET in China and 20.9 MJ/kg PET in Europe, with corresponding GWP values of 1.42 and 1.17 kg $CO_{2-eq}$/kg PET, respectively (Supplementary Tables S14 and S15). By comparison, conventional hydrolysis exhibits NREU and GWP values of 51 MJ/kg PET and 4.47 kg $CO_{2-eq}$/kg PET, respectively. Thus, the carbonylolysis route achieves reductions of 28.1 MJ/kg in NREU and 3.05 kg $CO_{2-eq}$/kg in GWP. These environmental advantages are primarily attributed to the exothermic nature of the carbonylation reaction, which thermodynamically drives PET depolymerization while minimizing external energy input. In addition, the integrated valorization of EG into PA eliminates the need for EG purification, thereby simplifying downstream processing and reducing overall process complexity.

Economic analysis from Aspen Process Economic Analyzer in Aspen Plus V11 (Fig. 5d and Supplementary Table S16) illustrated the distribution of costs including plant cost (installed cost, equipment cost, and capital cost), working costs (utilities cost and operating cost), raw materials costs (market for waste PET chips and CO), and product sales (market for TPA and PA). The total plant cost was USD 15.03 million, and the annual working cost was USD 3.1 million/year at an annual treatment of 100,000 tons waste PET chips. While raw materials represented the major cost component (USD 55.41 million/year), substantial product sales (USD 109.46 million/year) offset these expenses. The process demonstrates favorable economic viability with an annual profit of USD 35.92 million[27] and represents a promising return on investment for industrial implementation.

Together, the LCA and TEA highlight the carbonylolysis strategy as a promising and scalable platform for industrial PET recycling. This approach achieves high selectivity and atom economy, while minimizing environmental impacts and operational costs. Importantly, the strategy aligns with global sustainability objectives by integrating carbon efficiency, resource recovery, and energy minimization. These attributes make carbonylolysis particularly well-suited for regional industrial symbiosis, waste-to-chemicals initiatives, and petrochemical decarbonization efforts. These features position it as a compelling solution for large-scale, circular plastic valorization.

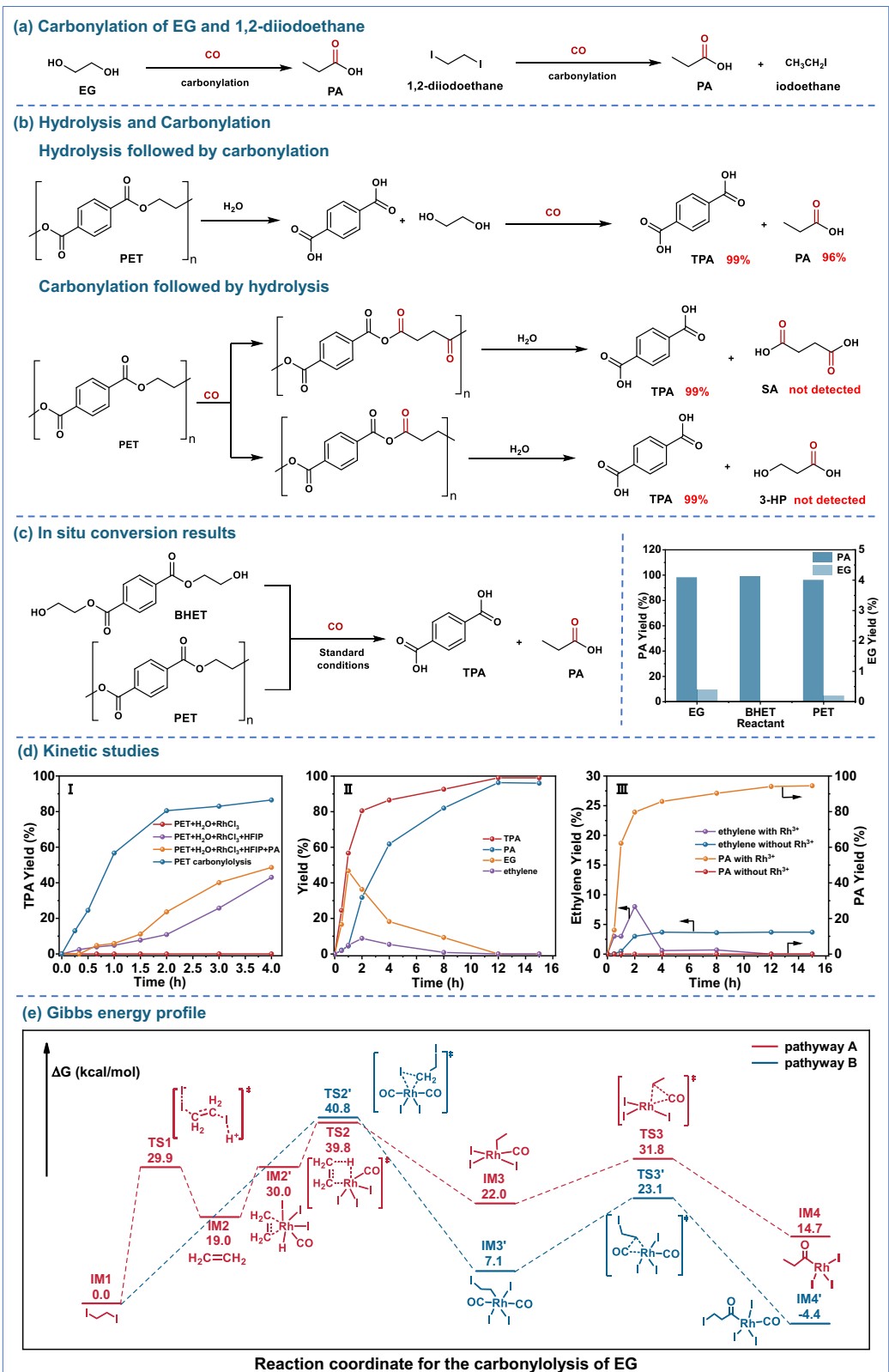

**Fig. 4 | Reaction pathway and mechanistic studies of PET carbonylolysis.**
**a** Carbonylation pathway of EG and 1,2-diiodoethane. **b** Products derived from different reaction sequences: (I) hydrolysis followed by carbonylation; (II) carbonylation followed by hydrolysis. **c** Carbonylolysis products and yields of BHET or PET. Standard reaction conditions: monomer (1 mmol structural unit) HFIP (2 mL), $H_2O$ (0.5 mL), $CH_3I$ (0.5 mmol), CO (2 MPa) and catalyst (RhCl₃, 5 mol%) at 170 °C for

12 h. **d** Kinetic studies: (I) time profiles of different reaction systems: PET (0.192 g, 1 mmol structural unit), HFIP (2 mL), $H_2O$ (0.5 mL), RhCl₃ (5 mol%), PA (50 µL). (II) kinetic profile of PET carbonylolysis; (III) time-dependent ethylene and PA evolution during EG carbonylation with and without $Rh^{3+}$. **e** Gibbs free-energy profile for the carbonylation of 1,2-diiodoethane.

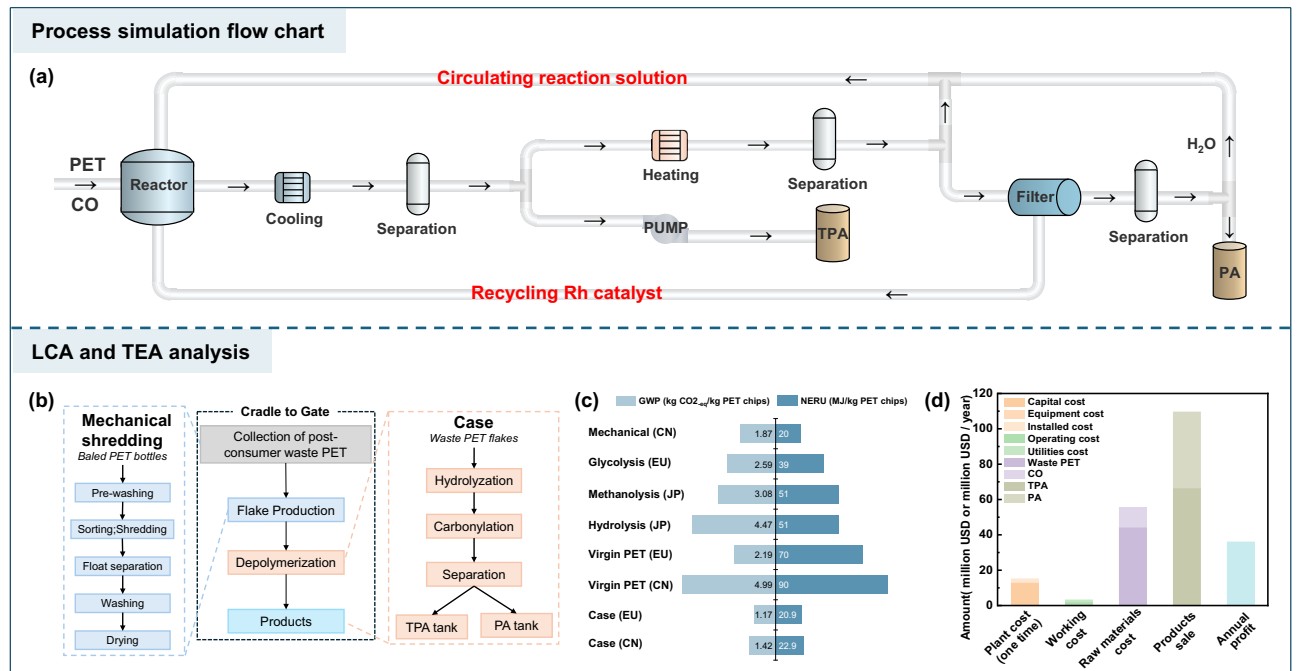

**Fig. 5 | Process flow of the PET carbonylolysis system with LCA and TEA.** **a** Process simulation flow chart for the carbonyloysis of waste PET. **b** Cradle-to-factory gate system boundary of closed-loop upcycling through PET carbonylolysis process based on the cut-off approach (illustrated by taking waste PET bottles as an example). **c** Sustainability metrics comparison: NREU and GWP[60,61,63]. **d** Analysis of cost contribution, product sales, and annual profit. CN (China), EU (European Union), and JP (Japan). Raw material costs and product pricing from databases (https://www.chemicalbook.com; https://www.100ppi.com) and literature sources.

## Discussion

The escalating global plastic waste crisis demands recycling solutions that are both technologically effective and environmentally sustainable. We present a one-step carbonylolysis strategy that simultaneously depolymerizes waste polyesters (e.g., PET) and reconstructs carbon chains in situ to yield two high-value organic acids (e.g., TPA and PA), using CO as a sustainable carbon feedstock. The key innovation lies in seamlessly coupling depolymerization with carbonylation, eliminating the diol separation step that typically constrains conventional recovery processes and avoiding carboxylic salt formation under basic conditions. This integration reduces acid/base consumption and wastewater generation, while also lowering the energy required for product purification. In addition, the exothermic nature of carbonylation enables partial self-heating, further reducing external energy demand. For PET, near-complete depolymerization was achieved under mild conditions, affording TPA and PA in 99% and 96% yields, respectively. Mechanistic investigations combining kinetic analysis, control experiments, and DFT calculations revealed a cascade sequence: ethylene glycol undergoes iodide-mediated di-substitution to 1,2-diiiodoethane, followed by elimination to ethylene and Rh-catalyzed carbonylation to PA. This pathway demonstrates a fundamentally new approach to reconfiguring carbon chains from plastic-derived diols. The carbonylolysis platform showed broad applicability, efficiently processing diverse polyesters (PBT, PEF) and complex real-world plastic waste streams. Techno-economic analysis projects an annual profit of USD 35.92 million for a 100,000-ton PET chip treatment capacity. Life cycle assessment shows substantial environmental benefits over existing chemical recycling, including reduced non-renewable energy use (22.9 and 20.9 MJ/kg PET for China and Europe) and lower global warming potential (1.42 and 1.17 kg $CO_{2\text{-eq}}$/kg PET). While promising, large-scale deployment requires advances in cost-efficient non-noble catalysts and robust performance with highly mixed or contaminated feedstocks, with catalyst recovery and operational efficiency being particularly critical. The one-step carbonylolysis

strategy establishes a transformative chemical paradigm for sustainable plastic upcycling: integrating polymer depolymerization with precise carbon-chain reconstruction to convert waste polyesters and sustainable CO into high-value products. By integrating waste-derived carbon and renewable CO within a single catalytic cycle, this approach pioneers innovative valorization pathways, advances circular-carbon science, decreases reliance on fossil feedstocks, and supports critical climate and resource-efficiency goals.

## Methods
### Catalytic tests
In a typical experiment, a mixture consisting of 0.192 g (1 mmol structural unit) PET powder, 500 μL RhCl₃ aqueous solution (0.05 mmol of $Rh^{3+}$), 0.071 g (0.5 mmol) CH₃I and 2 mL of HFIP were introduced into the reactor. The autoclave and container were then sealed and purged three times with CO to remove air, and then charged with 2.0 MPa CO. The reactor was then heated to 170 °C, -controlled by a temperature control device, and stirred at a speed of 700 rpm. After reaching the desired reaction time, the reactor was immediately quenched in cold water. Gas products were collected by an airbag and analyzed using an Agilent 6890 N gas chromatography equipped with an Agilent GS-Q GC Column. The liquid phase was diluted with 8 mL reaction solvent. Quantitative analysis of PA and EG in the liquid phase is performed using Shimadzu GC-2014 gas chromatography equipped with Shimadzu SH-WAX Polyethylene glycol GC Column, and TPA is performed using Bruker Avance neo (¹H at 500 MHz) Nuclear magnetic resonance (NMR). Chloroethane was selected as the gas internal standard. A mixture of 100 μL chloroethane and 400 μL reaction gas was identified by GC to quantify the yield of ethylene. To the reaction suspension, 1.0 mmol mesitylene was added as an internal standard. The suspensions were centrifuged at 2555 × g to separate the solid and liquid phases, and the supernatant was taken for product identification by GC to quantify the conversion rate of PA. The remaining liquid was vaporized to remove the solvent HFIP,

1 mmol of fumaric acid was added as an internal standard, and the suspension was centrifuged at 2555 × g to separate the solid and liquid phases. 50 μL of the supernatant was mixed with 550 μL D$_2$O for NMR analysis to quantify the conversion rate of TPA.

## Computational methods

All calculations were carried out using the Gaussian 16 package[58]. Both geometry optimizations and single-point energy calculations were optimized using the hybrid M06-2X functionals in combination with the def2-TZVP basis set for C, H, O, and the SDD basis set for I, Rh. Frequency analyses were conducted at the same level of theory to obtain thermal correction and confirm the stationary points to be minimal or transition states. The solvation effect of HFIP[46] and H$_2$O was introduced using the SMD model[59]. All the species involved were treated in standard conditions.

## Life cycle assessment and techno-economic analysis

LCA was conducted using Open LCA following the ISO 14040 series standard. The non-renewable energy use (NREU) and global warming potential (GWP) of our carbonylolysis versus currently reported PET recycling routes[60–62] were assessed. The functional unit is 1 kg of amorphous PET resin in chips form. The system boundary is "cradle-to-gate", which includes: the collection and transportation of waste PET, the production of PET flakes, and the carbonylolysis process to obtain TPA and PA (Fig. 5a, b). The background processes were modeled using generic datasets from the Ecoinvent v3.9.1 database, primarily based on globally averaged values. The CML-IA method was employed for NREU (expressed in megajoules, MJ), while the IPCC 2013 methodology was used for GWP (expressed in kilograms of CO$_2$-equivalents, kg CO$_{2\text{-eq}}$), both evaluated over a 100-year time horizon.

The experimental procedure under scrutiny was diligently formulated and modeled via the chemical process simulation software Aspen Plus V11, with a comprehensive schematic illustrated in the Supplementary Information. A cost-revenue analysis of 100,000-ton waste PET per year was performed using the established techno-economic assessment model. The summary for the estimated parameters and total investment cost included raw materials, plant working, plant construction, and product sales. The cost of raw materials and product sales was calculated from the elemental balance and unit price. The cost of plant working (including operating costs and utilities) was directly derived from Aspen Plus V11. Detailed data and assumptions can be found in the Supplementary Information. In this process simulation, the ASPEN simulation used the NRTL physical property method. The process commenced by introducing PET waste (WAS-PET, 12500 kg/h) mixed with water, HFIP, HI and RhCl$_3$ catalyst, and subsequently introduced into the reactor with CO. The depolymerization reaction was conducted at 170 °C under precise control.

## Data availability

The main data supporting the findings in this study are provided in the paper and Supplementary information. All data are available from the corresponding authors upon request. Source data are provided in this paper.

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

## Acknowledgements

This work was supported by the National Natural Science Foundation of China (22376183 to Q. M.), the Key Research and Development Program of Zhejiang Province (2024C03112 to Q. M.) and Zhejiang Provincial Outstanding Youth Science Foundation (LR26B070001 to Q. M.). We thank Dr. Jing Li for her technical assistance on the NMR characterization.

## Author contributions

D.L.: experiments perform, formal analysis, and writing-original draft. S.Z.: formal analysis. Q.M.: conceptualization, formal analysis, funding acquisition, resources, supervision and writing.

## Competing interests

Q.M. and D.L. are co-inventors on a patent application to be filed covering the polymer carbonylolysis pathway described in this work. All other authors declare no competing interests.
