## [Transparent Peer Review file · Nature Communications]

Carbonylolyis of waste polyesters into high-value organic acids

Corresponding Author: Professor Qingqing Mei

Version 0:

Reviewer comments:

Reviewer #1

(Remarks to the Author)

Liu et al. present a very interesting study on the valorization of PET using a rhodium-based carbonylation catalyst. Compared to existing approaches for PET valorization, they do not rely on hydrolysis of the polymer backbone but instead use CO to transform the C2 building block ethylene glycol into the C3 molecule propionic acid.

The study is well designed and executed, encompassing not only thorough chemical analysis but also an environmental assessment and a preliminary outline for technological implementation.

Overall, I am supportive of the publication of this article in Nature Communications after the authors conduct several additional experiments and provide clarification on specific aspects of the manuscript.

Major questions

- Role of CH₃I loading. Why is such a large excess of CH₃I required relative to the catalyst? In the classical Cativa process, iodine functions as a co-catalyst. Here, a distinct change in reactivity is observed between 0.1 and 0.3 equivalents of CH₃I, yet no explanation is provided for this nearly quantitative shift from EG to PA.
- Stability of HFIP under reaction conditions. How stable is HFIP in the presence of the Rh catalyst and CO? Are any fluorine species formed? A control experiment with Rh³⁺, HFIP, and CO in the absence of PET would be useful. ¹⁹F NMR may reveal small signals compared to the dominant HFIP resonance; in this case, an adapted solvent suppression sequence should be employed.
- Counter ions in metal salts. The Supporting Information lists the metal salts as chlorides. This information should be included in the main text. In this context, I am curious about the role of chloride: how does the reaction perform if RhI₃ is used directly, without an additional iodine additive?
- Figure 4D (b & c). The time axis presentation is confusing. Either equal step/tick distances or a logarithmic scale should be used. In panel (c), arrows are needed to link curves to the correct y-axes. Furthermore, the secondary axis showing "120% yield" is misleading — yields should be limited to 100%.
- Preparation of PET powder. How was the PET powder obtained? Was it ball-milled? Pretreatment methods such as ball milling can significantly affect polymer structure; see, for example, Vollmer et al. (10.1039/D4MR00098F; 10.26434/chemrxiv-2025-nkf6j). The method should be clearly described.
- Supporting Information. The SI should include at least one set of representative GC chromatograms from a catalytic test, to allow readers to assess the raw data quality and the data-processing workflow. It would be even better if the entire dataset is made available together with the analysis script.
- Reaction equipment. The autoclave manufacturer, type, and total volume must be specified.

Minor comments and clarifications

- Mechanistic statement. The manuscript states: "The presence of iodide was essential, as CH₃I reacts with Rh to form the catalytically active intermediate [Rh(CO)₂I₂]-." However, Reference 36 shows that CH₃I reacts with [Rh(CO)₂I₂]⁻ to form [Rh(CO)₂I₃(CH₃)]⁻. The statement should be revised to reflect the actual species involved.
- Environmental analysis. The sentence "Compared with conventional hydrolysis, the carbonylolytic route reduced 28.1 MJ/kg in NREU and 3.05 kg CO₂-eq/kg in GWP, respectively" is unclear. Are the numbers linked to the reference hydrolysis process. Please rephrase for clarity.
- Figure S5. The broad signals need to be assigned. Presumably, they correspond to water and the carboxylic acid proton?
- Scheme 1B. What does "R" represent?
- Figure 4E. Pathways A and B need to be explained explicitly.
- Competing interests. The authors declare no competing interests. Is this accurate? Was no patent application filed for this process?

Typos

- p. 4, l. 87: "mental catalyst" → "metal catalyst."

Reviewer #2

(Remarks to the Author)

This manuscript reports a one-pot "carbonylolytic" strategy that couples polyester depolymerization with in-situ carbon-chain reconstruction, producing high-value C₃+ carboxylic acids under relatively mild conditions. The experimental methodology is systematic. I believe it meets the publication standards of Nature Communication and can be published after the authors addressed the following comments.

My comments are as follows:

1. Section 2.1: The authors selected HFIP as the solvent due to its favorable solubility for PET. However, the manuscript does not address the potential use of alternative bio-based solvents, such as dimethyl isosorbide (DMI), which are more environmentally friendly and also exhibit good solubility for polyester materials. It is recommended that the authors clarify whether such green solvents were considered, or explain why they may not be suitable for the present reaction system.
2. Figure 2: The condition optimization experiments are presented without error bars. It is unclear whether the reported data represent results from a single experiment or the average of multiple replicates. If the latter, the corresponding error bars should be provided.
3. Section 2.4: The authors propose a catalyst separation and recovery strategy, but no experimental data are provided to validate its effectiveness. It is necessary to clarify whether catalyst recycling experiments were conducted. In particular, evidence is required to demonstrate that Rh can be completely recovered and that the recovered Rh retains comparable activity and selectivity in subsequent reaction cycles.
4. Section 2.4: CO is employed as a reactant in the carbonylolytic process. However, CO is also a pollutant. It is recommended that the authors clarify why the energy consumption and emissions associated with CO production were not included in the LCA. As this omission may significantly affect the reliability of the LCA results.
5. Figure 5: The authors mention circulating reaction solution, but no experimental evidence is provided to validate the feasibility of multiple reuse cycles. It is recommended that the authors provide data on solvent recovery efficiency and the purity after repeated use to demonstrate the practicality of solvent recycling.

Reviewer #3

(Remarks to the Author)

The authors report a one-pot "carbonylolytic" strategy to couple polyester depolymerization with in-situ carbon-chain reconstruction, producing high-value C₃ carboxylic acids under relatively mild conditions (170°C, 2 MPa 15 CO), which is indeed an innovative method. However, there are some major problems that haven't been addressed. First, Rh is a very expensive metal, so that when using it as a catalyst, it needs to have high TON or high stability to be recycled. Here, the author proposed to separate the Rh from the reaction system by converting it to inert RhI₃. It seems unpractical for actual application. Besides, more details need to be provided, such as the recovery percentage and the reactivation pathway of the catalyst. Second, iodide reagent is also kind of expensive, the author needs to address how to separate and recycle the iodide additive. Third, the author used hexafluoroisopropanol [HFIP] as solvent for its good properties. However, it is an expensive solvent and has serious environmental impact, so that its recovery needs to be addressed. Fourth, the author claims that the carbonylation mechanism is through the ethylene insertion into [Rh(CO)₂I₂]⁻ active catalytic species. Actually, the well-recognized active catalytic species is supposed to be [HRh(CO)I₃]⁻. You can follow a recently published paper J. Am. Chem. Soc. 2025, 147, 38, 34589–34600.

Version 1:

Reviewer comments:

Reviewer #1

(Remarks to the Author)

After reviewing the revisions made by Liu et al., I am fully supportive of the publication of the manuscript in Nature Communications. The authors have fully addressed the three reviewers' concerns by not only adding additional clarifications where needed, but by also providing additional experimental data which support their claims made throughout the manuscript.

Reviewer #2

(Remarks to the Author)

This manuscript can be accepted as it is.

Reviewer #3

(Remarks to the Author)

The author has answered most of the questions. But I think the answers to the reviewer's comments are not sufficient, the following issues are still need to be addressed.

1. The author claimed in the response that "A sharp increase in activity was observed upon increasing CH₃I loading from sub-threshold to iodide-rich conditions, followed by a plateau upon moderate further increases, indicating that a minimum iodide level is required to establish the active Rh catalytic cycle rather than a linear dependence on CH₃I loading." However, only three data provided in not enough to draw this conclusion. I think more data should be added to draw a relationship line. Besides, the author should use this data to correlate to the catalytically active Rh-iodide species and give a reasonable explanation.

2. The author mentioned in the response that "Notably, all PET samples were manually cut into small pieces without further powdering or micronization, thereby minimizing potential structural modification associated with intensive mechanical pretreatments such as ball milling." Is there any study on the effect of the PET sample size? It seems unreasonable that the result is irrelevant to the substrate size.

3. The author described a complex procedure for the catalyst recycling. First, only 95% of Rh can be recycled, which would lead to significant high cost. A reasonable production cost needs to be provided. Second, the recycling efficiency of iodide species should be calculated, which also contribute to the high cost of the procedure.

Version 2:

Reviewer comments:

Reviewer #3

(Remarks to the Author)

I think all the questions have been answered, it is ready to be accepted

Reviewer #1 (Remarks to the Author):

Liu et al. present a very interesting study on the valorization of PET using a rhodium-based carbonylation catalyst. Compared to existing approaches for PET valorization, they do not rely on hydrolysis of the polymer backbone but instead use CO to transform the C₂ building block ethylene glycol into the C₃ molecule propionic acid.

The study is well designed and executed, encompassing not only thorough chemical analysis but also an environmental assessment and a preliminary outline for technological implementation.

Overall, I am supportive of the publication of this article in Nature Communications after the authors conduct several additional experiments and provide clarification on specific aspects of the manuscript.

Response: We appreciate the reviewer for the positive comments and suggestions for improving this manuscript.

Major questions

Comments 1: Role of CH₃I loading. Why is such a large excess of CH₃I required relative to the catalyst? In the classical Cativa process, iodine functions as a co-catalyst. Here, a distinct change in reactivity is observed between 0.1 and 0.3 equivalents of CH₃I, yet no explanation is provided for this nearly quantitative shift from EG to PA.

Response: We thank the reviewers for their insightful comments regarding the role and loading of CH₃I in the catalytic system. In Rh-catalyzed carbonylation chemistry, an excess iodide source is widely recognized as essential for sustaining catalytic activity through control of Rh speciation. In the present system, CH₃I functions as both the iodide source and a co-catalyst, governing the formation and stability of the catalytically competent Rh–iodide carbonyl species.

As established in classical Monsanto/Cativa-type carbonylation chemistry, oxidative addition of CH₃I to Rh constitutes a key elementary step, and the steady-state population of active Rh–iodide species is highly sensitive to iodide activity.

Consequently, a sufficient excess of CH₃I is required to maintain an iodide-rich environment that stabilizes the active Rh intermediates necessary for efficient carbonylation (*Sci. Adv.*, **2018**, 4, eaaq0266; *Green Chem.*, **2019**, 21, 4434–4442; *Angew. Chem. Int. Ed.*, **2017**, 56, 14868–14872; *ACS Catal.*, **2021**, 11, 7249–7256; *Green Chem.*, **2024**, 26, 7302–7311).

The abrupt change in reactivity observed between 0.1 and 0.3 equivalents of CH₃I reflects a speciation-controlled threshold, rather than a linear concentration effect. At low CH₃I loadings (e.g., 0.1 equiv), iodide activity is insufficient to sustain the catalytically active Rh–iodide species, leading to incomplete carbonylation and predominant formation of EG. Once the iodide concentration exceeds a critical level, the catalytically competent Rh-iodide cycle becomes fully established, resulting in near-quantitative conversion to PA.

To further support this interpretation, we examined an intermediate CH₃I loading (0.2 equiv), which afforded a TPA yield of 99% and a PA yield of 83%, with no detectable EG formation. These results are comparable to those obtained at 0.3 equivalents of CH₃I, indicating that once the critical iodide threshold is exceeded, further increases in CH₃I loading do not lead to proportional enhancements in activity. This behavior is consistent with a speciation-controlled catalytic regime rather than a stoichiometric dependence on CH₃I.

Revisions made: The manuscript has been revised to clarify the role of CH₃I loading: *“The presence of iodide was essential for carbonylation activity, as a sufficiently high iodide concentration is required to stabilize the catalytically active Rh–iodide carbonyl species. At low CH₃I loadings, iodide activity was insufficient to sustain the active Rh speciation, resulting in incomplete carbonylation and predominant formation of EG. A sharp increase in activity was observed upon increasing CH₃I loading from sub-threshold to iodide-rich conditions, followed by a plateau upon moderate further increases, indicating that a minimum iodide level is required to establish the active Rh catalytic cycle rather than a linear dependence on CH₃I loading. At higher CH₃I loadings, excessive iodide led to reduced PA yields, likely due to over-stabilization of Rh–iodide complexes and diminished catalytic turnover (Fig. 2A).”*

Comments 2: Stability of HFIP under reaction conditions. How stable is HFIP in the presence of the Rh catalyst and CO? Are any fluorine species formed? A control experiment with Rh³⁺, HFIP, and CO in the absence of PET would be useful. ¹⁹F NMR may reveal small signals compared to the dominant HFIP resonance; in this case, an adapted solvent suppression sequence should be employed.

Response: We thank the reviewer for the helpful comments regarding the stability of HFIP, which is indeed critical for assessing the robustness of the reaction system.

Following these suggestions, we conducted a series of control experiments to evaluate HFIP stability. First, reactions were carried out in the absence of PET but under otherwise identical conditions (Rh, HFIP, CH₃I, and CO at the same temperature and pressure). Second, both PET and iodide additives were omitted, leaving only Rh, HFIP, and CO under identical conditions. In both cases, ¹⁹F NMR spectra analysis showed no detectable changes in the dominant HFIP resonance and no additional fluorine-containing signals, indicating that HFIP remains chemically stable in the presence of Rh and CO. These spectra were acquired with appropriate signal suppression to ensure that minor fluorinated species, if present, would be observable. Furthermore, solvent recycling experiments demonstrated that HFIP can be reused multiple times without loss of catalytic performance, further confirming its stability under the reaction conditions.

Revisions made: In the revised Supplementary Information, the corresponding ¹⁹F NMR data (Figure S7) and recycling experiments have been added to explicitly demonstrate the chemical integrity and reusability of HFIP under Rh/CO conditions (see also our response to Reviewer 2, Comment 5).

Figure R1. ^{19}F NMR spectra (D_2O) of the reaction solutions: (A) without PET; (B) without PET and CH_3I ; (C) HFIP standard. Corresponding data have been added as Figure S7 in the revised Supplementary Information.

Comments 3: Counter ions in metal salts. The Supplementary Information lists the metal salts as chlorides. This information should be included in the main text. In this context, I am curious about the role of chloride: how does the reaction perform if RhI_3 is used directly, without an additional iodine additive?

Response: We thank the reviewer for the insightful comments. Following your suggestion, we have now explicitly stated in the main text that all metal salt catalysts used in this study were chloride salts.

Under the iodide-rich carbonylation conditions employed here, chloride is not mechanistically essential and is rapidly displaced to form the catalytically active Rh/iodide species, consistent with established Monsanto/Cativa-type systems (*Organometallics*, **1994**, 13, 3215–3226; *Coord. Chem. Rev.*, **2003**, 243, 125–142; *J. Am. Chem. Soc.*, **2022**, 144, 2034–2050). To examine whether chloride influences catalytic performance, we conducted a series of control experiments (Table R1). When RhI_3 was used in place of RhCl_3 in combination with the HI promoter, full catalytic activity comparable to the standard system was observed, attributable to rapid in situ

anion exchange generating the active Rh–iodide carbonyl species. However, when RhI_3 was used as the sole rhodium precursor without any additional iodide additive, only low activity (13%) was observed. In contrast, when RhCl_3 was used as the sole rhodium precursor, only trace formation of PA was detected. These results confirm that chloride is not required for catalytic activity and does not provide any promotional effect. Instead, they indicate that the iodide supplied stoichiometrically by RhI_3 is insufficient to establish the high free-iodide activity required to promote oxidative addition, stabilize key acyl-forming intermediates, and sustain the Cativa/Monsanto-type catalytic cycle.

Collectively, these findings demonstrate that catalytic efficiency is governed by the concentration of free iodide rather than the identity of the Cl^- counter-anion, and that maintaining a sufficiently high iodide activity is essential for efficient catalytic turnover.

Revisions made: In the revised manuscript, the catalysts (Rh^{3+} , Ru^{3+} , Ir^{3+} , Pd^{2+} , Co^{2+} , Ni^{2+}) have been updated to their corresponding chloride salts (RhCl_3 , RuCl_3 , IrCl_3 , PdCl_2 , CoCl_2 , NiCl_2), and the statement “the effect of the chloride counter-anion was negligible (Table S1)” has been added. In the revised Supplementary Information, Table S1 has been included.

Table R1. Catalytic performance of different anionic rhodium salts. (**Reproduced as Table S1 in the revised Supplementary Information**)

No.	Rh	HI	TPA Yield (%)	PA Yield (%)	EG Yield (%)
1	RhI_3	HI	>99	97.6	0.0
2	RhI_3	-	97.4	13.0	83.8
3	RhCl_3	HI	>99	93.0	0.0
4	RhCl_3	-	93.6	0.1	76.7

Standard reaction conditions: PET (0.192g, 1 mmol structural unit), HFIP (2 mL), H_2O (0.5 mL), HI (0.4 mmol), CO (2 MPa) and catalyst (RhCl_3 or RhI_3 , 5 mol%) at 170°C for 12 h.

Comments 4: Figure 4D (b & c). The time axis presentation is confusing. Either equal step/tick distances or a logarithmic scale should be used. In panel (c), arrows are needed to link curves to the correct y-axes. Furthermore, the secondary axis showing “120% yield” is misleading — yields should be limited to 100%.

Response: We thank the reviewer for the constructive suggestions regarding the presentation of Figure 4D (b & c). We agree that the previous time-axis formatting could be confusing, and the panels have now been revised to use uniform tick spacing, ensuring consistent visual interpretation of the kinetic profiles. In panel (c), arrows have been added to clearly associate each curve with its corresponding y-axis, and the dual-axis labeling has been redrawn for improved clarity. In addition, the secondary axis previously extending beyond 100% yield has been corrected, as yields should not exceed 100%. The revised figure now limits the yield scale to 0–100% and presents the kinetic and selectivity data in a clearer and more accurate manner.

Revisions made: In the revised manuscript, Figure 4D (b & c) has been updated.

Figure R2. Revised Figure 4D (b & c) of the manuscript. Kinetic studies: (a) time profiles of different reaction systems: PET (0.192 g, 1 mmol structural unit), HFIP (2 mL), H₂O (0.5 mL), RhCl₃ (5 mol%), PA (50 μ L). (b) kinetic profile of PET carbonyllysis; (c) time-dependent ethylene and PA evolution during EG carbonylation with and without Rh³⁺.

Comments 5: Preparation of PET powder. How was the PET powder obtained? Was it ball-milled? Pretreatment methods such as ball milling can significantly affect polymer structure; see, for example, Vollmer et al. (10.1039/D4MR00098F; 10.26434/chemrxiv-2025-nkf6j). The method should be clearly described.

Response: We thank the reviewer for highlighting this important point.

The PET powder used in this study ($M_n \approx 150,000$) was commercially supplied by Tesulang Chemical Materials Co., Ltd. According to the supplier specifications, the material is produced via conventional industrial size-reduction processes (low-shear-pulverization) designed for polymer handling, rather than mechanochemical milling. It was used as received and not subjected to any high-energy mechanochemical pretreatment (e.g., ball milling), and is therefore not expected to induce significant chain scission or structural modification.

We fully agree that high-energy ball milling and related mechanochemical pretreatments can substantially alter polymer structure, as demonstrated in the studies cited by the reviewer. However, PET powder preparation is not a prerequisite for this transformation. As shown in Figure 3A, real-world waste PET samples in various practical physical forms were manually cut into small pieces without further micronization or powdering. This simple treatment minimizes potential structural modification induced by mechanical pretreatment. The transformation performance observed across these PET forms is comparable, demonstrating that the process is tolerant to realistic PET morphologies and enabling direct application to real plastic waste streams.

Revisions made: In the revised manuscript, the sentence *“We systematically optimized key parameters to enhance PA yield”* has been revised to *“Using commercially sourced, low-shear-pulverized PET powder as the feedstock (without additional ball-milling pretreatment that could alter the polymer structure), we systematically optimized key parameters to maximize PA yield.”* We also added: *“Notably, all PET samples were manually cut into small pieces without further powdering or micronization, thereby minimizing potential structural modification associated with intensive mechanical pretreatments such as ball milling. These results demonstrate that PET powder preparation is not a prerequisite for the transformation, enabling direct application to real waste streams without energy-intensive size reduction.”* (p. 4, line 12) **In the revised Supplementary Information:** additional information on the PET source and handling has been added.

Comments 6: Supplementary Information. The SI should include at least one set of representative GC chromatograms from a catalytic test, to allow readers to assess the raw data quality and the data-processing workflow. It would be even better if the entire dataset is made available together with the analysis script.

Response: We thank the reviewer for the constructive suggestion regarding data transparency and evaluation of the GC analysis. In response, representative GC chromatograms from typical catalytic experiments have been added to the Supplementary Information, displaying the raw detector signals together with the corresponding peak assignments used for quantification. In addition, the peak integration and yield calculation has been provided, enabling readers to assess both the chromatographic data quality and the analysis workflow. Additional raw GC files are available from the authors upon reasonable request.

Revisions made: Representative GC chromatograms and the associated data-processing procedures have been added to the revised Supplementary Information (Section 2.6, *Quantification of Product Yields*, Fig. S6-I–III).

Figure R3. (A) representative GC chromatograms from a typical catalytic run; (B) Calibration curve of PA using mesitylene as the internal standard; (C) Calibration curve of EG using mesitylene as the internal standard.

Comments 7: Reaction equipment. The autoclave manufacturer, type, and total volume must be specified.

Response: We thank the reviewer for the suggestion to enhance the experiment details.

In response, the Experimental section has been revised to explicitly describe the reaction apparatus. All carbonylolytic reactions were conducted in a stainless-steel stirred autoclave manufactured by Ruisheng Laboratory Equipment Factory (Lüshunkou District, Dalian). The reactor is a high-pressure-resistant steel vessel equipped with a PTFE liner. The total internal volume of the autoclave was determined to be 24 mL, and the usable volume of the PTFE liner was 19.4 mL. The reactor was purged and pressurized with CO following the procedure described in the manuscript.

Revisions made: In the revised **Supplementary Information**, the manufacturer, reactor type, and total and usable volumes of the autoclave have been added to Section 1.2, *Reactor Equipment*, with corresponding photos shown in Fig. S1.

Figure R4. Details of the reaction equipment. Photograph of the high-pressure reactor (A); Cutaway view of the high-pressure reactor (B); Cross-sectional view of the reactor

liner (C); Side view of the reactor liner (D).

Minor comments and clarifications

Comments 8: Mechanistic statement. The manuscript states: " The presence of iodide was essential, as CH₃I reacts with Rh to form the catalytically active intermediate [Rh(CO)₂I₂]⁻." However, Reference 36 shows that CH₃I reacts with [Rh(CO)₂I₂]⁻ to form [Rh(CO)₂I₃(CH₃)]⁻. The statement should be revised to reflect the actual species involved.

Response: We thank the reviewer for pointing out the inaccurate mechanistic wording regarding the role of CH₃I. We agree that CH₃I does not directly react with Rh to form the anionic Rh(I) species [Rh(CO)₂I₂]⁻. Instead, as established in Rh-catalyzed carbonylation chemistry, CH₃I primarily serves as an iodide promoter, while oxidative addition of CH₃I to [Rh(CO)₂I₂]⁻ can generate a transient methyl–Rh intermediate [(CH₃)Rh(CO)₂I₃]⁻, as described in Reference 36. This intermediate is short-lived and does not represent the catalytically dominant species.

Accordingly, the manuscript has been revised to remove the incorrect causal statement and to clarify that the essential role of CH₃I is to maintain a sufficiently high iodide activity required to stabilize the anionic Rh(I) carbonyl species responsible for carbonylation catalysis. The corrected text no longer implies that CH₃I directly generates [Rh(CO)₂I₂]⁻. As multiple Rh-iodide species may be involved (discussed in the Mechanism section), individual Rh-iodide intermediates are not explicitly specified in this paragraph.

Revisions made: In the revised manuscript, The mechanistic wording in the optimization discussion has been corrected to accurately describe the role of iodide and CH₃I. (p. 2, line 18)

Comments 9: Environmental analysis. The sentence "Compared with conventional hydrolysis, the carbonylolytic route reduced 28.1 MJ/kg in NREU and 3.05 kg CO₂-eq/kg in GWP, respectively" is unclear. Are the numbers linked to the reference

hydrolysis process. Please rephrase for clarity.

Response: We thank the reviewer for pointing out the ambiguity. The sentence has been revised to explicitly state that the reported reductions in NREU and GWP are referenced to the conventional hydrolysis process. The revised text now clearly reports both the absolute values for the hydrolysis benchmark and the corresponding values for the carbonylolytic route, eliminating any ambiguity in the comparison. (p. 7, line 14)

Comments 10: Figure S5. The broad signals need to be assigned. Presumably, they correspond to water and the carboxylic acid proton?

Response: We thank the reviewer for this helpful observation regarding the broad signals in Figure S5. Accordingly, the broad resonance at $\delta \approx 3.5$ ppm has been assigned to residual water in HFIP, which typically appears as a broadened peak due to strong hydrogen bonding. The broader signal at $\delta \approx 8.0$ ppm is attributed to proton exchange involving trace water and HFIP rather than carboxylic acid proton.

Revisions made: In the revised **Supplementary Information**, the original Figure S5 has been renumbered as Figure S8, and the broad resonances have been assigned and annotated.

Comments 11: Scheme 1B. What does "R" represent?

Response: We thank the reviewer for pointing out the ambiguity.

Accordingly, Scheme 1B has been revised to include clear definitions of both R and m. In the updated scheme, R denotes the divalent organic residue of the diol monomer, which is the portion of the monomer between the two hydroxyl groups, and m represents the number of carbon atoms in this residue. These clarifications have been incorporated directly into the scheme.

Revisions made: In the revised **manuscript**, Scheme 1B has been updated to

explicitly define the symbols R and m for improved clarity.

Comments 12: Figure 4E. Pathways A and B need to be explained explicitly.

Response: We thank the reviewer for the insightful comment. In the revised manuscript, we have clarified Pathways A and B. As illustrated in Fig. 4E, both pathways originate from the common intermediate **IM1** and diverge via β -elimination (Pathway A) or direct oxidative addition (Pathway B). In Pathway A, **IM1** undergoes β -elimination to generate ethylene, which is subsequently carbonylated to form PA. In Pathway B, **IM1** directly coordinates to the catalyst and undergoes oxidative addition followed by carbonylation, potentially producing 3-iodopropionic acid and succinic acid. DFT calculations indicate that β -elimination has a substantially lower free-energy barrier (29.9 vs. 40.8 kcal/mol), consistent with the selective formation of PA observed experimentally. These clarifications are now explicitly included in the main text to remove any ambiguity regarding Fig. 4E.

Revisions made: In the revised manuscript, the mechanistic descriptions of Pathways A and B in Fig. 4E have been expanded and clarified in the main text to clearly convey their distinct sequences and outcomes (p. 6, line 9).

Comments 13: Competing interests. The authors declare no competing interests. Is this accurate? Was no patent application filed for this process?

Response: We thank the reviewer for raising this important point regarding the declaration of competing interests. We confirm that, at the time of the original manuscript submission, no patent application had been filed and the statement “The authors declare no competing interests” was accurate.

Following your comment and a further evaluation of the potential applicability of this work, we have now decided to pursue a patent application related to the reported process. To ensure full transparency, the Competing Interests statement has been updated accordingly.

Revisions made: In the revised manuscript, the Competing Interests section has

been revised to reflect the intention to file a patent application.

Typos

Comments 14: p. 4, l. 87: “mental catalyst” → “metal catalyst.”

Response: We thank the reviewer for pointing out this typographical error. “Mental catalyst” has been corrected to “metal catalyst” **in the revised manuscript.**

Reviewer #2 (Remarks to the Author):

This manuscript reports a one-pot “carbonylolysis” strategy that couples polyester depolymerization with in-situ carbon-chain reconstruction, producing high-value C3+ carboxylic acids under relatively mild conditions. The experimental methodology is systematic. I believe it meets the publication standards of Nature Communication and can be published after the authors addressed the following comments.

Response: We appreciate the reviewer for the positive comments and suggestions for improving this manuscript.

My comments are as follows:

Comments 1: Section 2.1: The authors selected HFIP as the solvent due to its favorable solubility for PET. However, the manuscript does not address the potential use of alternative bio-based solvents, such as dimethyl isosorbide (DMI), which are more environmentally friendly and also exhibit good solubility for polyester materials. It is recommended that the authors clarify whether such green solvents were considered, or explain why they may not be suitable for the present reaction system.

Response: We thank the reviewer for the valuable suggestion regarding the potential use of bio-based solvents in place of HFIP. We fully agree that the development of greener solvent systems is important and merits careful consideration.

Accordingly, we evaluated representative bio-based solvents, including dimethyl isosorbide (DMI) and γ -valerolactone (GVL), for PET carbonylolysis under otherwise identical conditions. As shown in Table R2, PET depolymerization was incomplete using DMI as the solvent, affording only 30.7% TPA, while GVL enabled efficient PET depolymerization (95.4% TPA). However, neither solvent led to detectable formation of PA, indicating that the carbonylolysis pathway was not operative in these media. These results suggest that, despite adequate PET solubility benefit for depolymerization, DMI and GVL do not support the key carbonylolysis steps required for PA formation.

This incompatibility can be attributed to the unique solution properties of HFIP. HFIP provides a strongly hydrogen-bond-donating, highly ionizing, and low-nucleophilicity environment that promotes HI dissociation, stabilizes iodide and protonated intermediates, and supports the formation and persistence of anionic Rh/I catalytic species. These combined effects generate a superacid-like medium that is critical for efficient C–O activation and subsequent carbonyl insertion steps (*Synth.*, **2007**, 2007, 2925–2943; *J. Am. Chem. Soc.*, **2006**, 128, 8421–8426). In contrast, DMI and GVL lack sufficient hydrogen-bond donation and ion-stabilization capacity to sustain the iodide activity and acidic conditions necessary for the iodine-mediated carbonylolytic sequence (*Synth.*, **2007**, 2007, 2925–2943). Therefore, despite their greener origin and ability to dissolve polyesters to enhance depolymerization, these solvents are intrinsically incompatible with the mechanistic requirements of the present integrated carbonylolytic process.

We fully agree that identifying greener solvent systems capable of meeting these mechanistic requirements is an important direction and is currently under exploration in our ongoing work.

Revisions made: In the revised manuscript, bio-based solvents (DMI and GVL) have been included in the solvent evaluation. (p. 2, line 22) **In the revised Supplementary Information**, the corresponding reaction results for these bio-based solvents are provided in Figure S4.

Table R2. Reaction performance of bio-based solvents. (These data have been added to Fig. S4)

No.	Sol.	TPA Yield (%)	EG Yield (%)	PA Yield (%)
1	DMI	30.7	26.2	-
2	γ -Valeroactone	95.4	18.0	-

Standard reaction conditions: PET (0.192g, 1 mmol structural unit), solvent (2 mL), H₂O (0.5 mL), CH₃I (0.5 mmol), CO (2 MPa) and catalyst (RhCl₃, 5 mol%) at 170°C for 12 h.

Comments 2: Figure 2: The condition optimization experiments are presented without error bars. It is unclear whether the reported data represent results from a single experiment or the average of multiple replicates. If the latter, the corresponding error bars should be provided.

Response: We thank the reviewer for pointing out the need for greater clarity regarding the data presented in Figure 2. The optimization data were obtained as averages from at least three parallel experimental replicates and showed good reproducibility with small standard deviations. Error bars were not included in the original submission to maintain visual clarity and facilitate comparison of overall trends across a broad parameter space. We fully agree that inclusion of error bars improves scientific rigor. Following your suggestion, we have now added the corresponding error bars (standard deviations) to Figure 2, and the figure caption has been revised accordingly.

Revision made: In the revised manuscript, Figure 2 and the caption has been updated.

Figure R6. Updated Figure 2 of the manuscript. Construction of the catalytic system and optimization of conditions. Reaction conditions: PET (0.192 g, 1 mmol structural unit), HFIP

(2 mL), H₂O (0.5 mL), CH₃I (0.5 mmol), CO (2 MPa) and catalyst (RhCl₃, 5 mol%) at 170°C for 12 h. (A) The amount of CH₃I; (B) Water volume; (C) CO pressure; (D) Temperature and (E) Reaction time.

Comments 3: Section 2.4: The authors propose a catalyst separation and recovery strategy, but no experimental data are provided to validate its effectiveness. It is necessary to clarify whether catalyst recycling experiments were conducted. In particular, evidence is required to demonstrate that Rh can be completely recovered and that the recovered Rh retains comparable activity and selectivity in subsequent reaction cycles.

Response: We thank the reviewer for the insightful comment regarding catalyst recovery and recyclability. We have conducted dedicated experiments to experimentally validate Rh catalyst recovery and reuse via an RhI₃-based route.

Using an RhI₃-based laboratory recovery protocol, a closed Rh mass balance was achieved within experimental uncertainty. The experimental details: After completion of the reaction, the crude mixture was centrifuged to separate the solid fraction (*Solid a*) from the liquid phase (*liquid a*). *Solid a* was washed with aqueous HI to dissolve any trace Rh precipitate that might be present, and the wash solution was collected as *liquid b*. *Liquid a* was distilled to recover the HI/HFIP mixture, which was directly reused in the repeated catalytic cycle. The remaining liquid was evaporated to recover PA. The final residue, containing Rh complexes, was combined with *Liquid b* (denoted as *Liquid c*) and thermally aged to promote precipitation of RhI₃. The mixture was subsequently centrifuged to isolate solid RhI₃.

Using this procedure, more than 95% of the total Rh was recovered, with the remaining loss attributed to unavoidable handling and analytical uncertainties inherent to small-scale experiments. Under standard reaction conditions, the recycled Rh catalyst (Rh-cyclic) exhibited activity and selectivity comparable to fresh RhCl₃ and simulated recycled RhI₃ (prepared by thermal aging of RhCl₃ in excess aqueous HI) when used with fresh HFIP and HI, confirming efficient reactivation and preservation of catalytic integrity (Table R3).

Revision made: Details of the Rh recovery procedure and recycling performance have been added to the revised Supplementary Information (Fig. S10 and Table S4).

Figure R7. Rhodium -catalyzed cyclic processes within the laboratory setting.

Table R3. Reaction performance of different anionic rhodium salts.

No.	Rh	I ⁻	TPA Yield (%)	PA Yield (%)	EG Yield (%)
1	RhCl ₃	CH ₃ I	>99	96.0	0.0
2	RhI ₃	HI	>99	97.6	0.0
3	Rh-cyclic	HI	>99	95.9	0.0

Standard reaction conditions: PET (0.192g, 1 mmol structural unit), HFIP (2 mL), H₂O (0.5 mL), CH₃I (0.5 mmol) or HI (0.4 mmol), CO (2 MPa) and catalyst (RhCl₃, 5 mol%) at 170°C for 12 h.

Comments 4: Section 2.4: CO is employed as a reactant in the carbonylolytic process. However, CO is also a pollutant. It is recommended that the authors clarify why the energy consumption and emissions associated with CO production were not included in the LCA. As this omission may significantly affect the reliability of the LCA results.

Response: We thank the reviewer for raising this important point regarding the treatment of CO in the LCA. We agree that the handling of CO-related energy consumption and emissions must be clearly defined to ensure the reliability and transparency of the results.

In our analysis, CO is treated as a reactant (raw material) rather than an emission

source. The environmental burdens associated with CO production were fully included through upstream inventory datasets incorporated into the LCA model (Figure 5D and Tables S12–S13).

Specifically, the CO consumed in this process was modeled using standard industrial carbon monoxide production datasets, which incorporates the energy input, resource consumption, and emissions associated with CO production (typically via syngas generation or partial oxidation routes). As a result, the environmental impacts of CO production are already embedded within the system boundary as part of the raw-material inputs, in accordance with ISO 14040/14044 guidelines (*Chem. Eng. J.*, **2024**, 489, 151326; *Green Chem.*, **2017**, 19, 2244–2259).

We acknowledge that CO is a regulated pollutant. However, in industrial practice, CO is highly reactive process gas and is routinely recovered and recycled in large-scale carbonylation and hydroformylation processes, such as Cativa/Monsanto-type systems, to minimize emissions and improve efficiency. A similar CO recycling strategy would be applicable to the present process, thereby preventing atmospheric release of unreacted CO.

Revision made: In the revised manuscript, we have revised that CO is treated as a purchased industrial feedstock (p. 7, line 1) and that its upstream environmental impacts are included in the LCA model (Figure 5D). **In the revised Supplementary Information**, the specific values are provided in Tables S12–S13.

Comments 5: Figure 5: The authors mention circulating reaction solution, but no experimental evidence is provided to validate the feasibility of multiple reuse cycles. It is recommended that the authors provide data on solvent recovery efficiency and the purity after repeated use to demonstrate the practicality of solvent recycling.

Response: We thank the reviewer for this constructive suggestion regarding the feasibility of circulating the reaction solution. Following your suggestion, additional experiments were conducted to evaluate the recovery efficiency, purity, and reusability of HFIP under the reaction conditions.

HFIP recycling procedure: After completion of the reaction, the crude mixture was centrifuged to separate the solid fraction from the liquid phase. The liquid phase was then subjected to low-temperature distillation to recover an HI/HFIP mixed solution, which was directly reused in subsequent catalytic cycles without further purification. Because HFIP and HI function cooperatively as the reaction medium, separation of HI from HFIP is neither required nor desirable for effective solvent circulation. Moreover, the co-recycling of solvent and iodide promoter is consistent with established industrial practice in Monsanto/Cativa-type carbonylation processes.

HFIP was recovered after each run with near-quantitative efficiency (>99%) and reused for at least three consecutive reaction cycles. ^1H , ^{13}C and ^{19}F NMR analyses of the recycled solvent showed no detectable impurity accumulation relative to fresh HFIP. Consistently, PET conversion and product selectivity remained essentially unchanged over multiple cycles with fresh RhCl_3 , indicating that solvent purity and catalytic performance were preserved upon reuse (Figures R8–R10, Table R4).

These results experimentally validate the feasibility of circulating the reaction solution and demonstrate that HFIP can be efficiently recovered and reused without loss of solvent purity or catalytic performance, validating the feasibility of circulating the reaction medium.

Revision made: In the revised Supplementary Information, the ^1H , ^{13}C and ^{19}F NMR spectra of the fresh HFIP standard and the recycled HFIP solution are provided in Fig. S11–S13, and the corresponding catalytic results are summarized in Tables S5.

Figure R8. ^1H -NMR (DMSO-d_6) of standard sample: (A) Fresh HFIP; (B) HFIP recycled three times.

Figure R9. ^{13}C -NMR (DMSO-d_6) of standard sample: (A) HFIP; (B) HFIP recycled three times.

(A) Hexafluoroisopropanol (HFIP)

(B) HFIP recycled three times

Figure R10. ^{19}F -NMR (D_2O) of standard sample: (A) HFIP; (B) HFIP recycled three times.

Table R4. Recyclability Test of HFIP

OC(=O)c1ccc(cc1)C(=O)OCC + CO >> OC(=O)c1ccc(cc1)C(=O)O + CC(=O)O

Cycle Number	Temp.	Sol.	TPA Yield (%)	PA Yield (%)
1	170°C	HFIP	99	96.3
2	170°C	HFIP	99	96.7
3	170°C	HFIP	99	96.4

Standard reaction conditions: PET (0.192g, 1 mmol structural unit), HFIP (2 mL), H_2O (0.5 mL), CH_3I (0.5 mmol), CO (2 MPa) and catalyst (RhCl_3 , 5 mol%) at 170°C for 12 h.

Reviewer #3 (Remarks to the Author):

The authors report a one-pot “carbonylolysis” strategy to couple polyester depolymerization with in-situ carbon-chain reconstruction, producing high-value C3 carboxylic acids under relatively mild conditions (170°C, 2 MPa CO), which is indeed an innovative method. However, there are some major problems that hasn't been addressed.

Response: We appreciate the reviewer for the positive comments and suggestions for improving this manuscript.

Comments 1: First, Rh is a very expensive metal, so that when using it as a catalyst, it needs to have high TON or high stability to be recycled. Here, the author proposed to separate the Rh from the reaction system by converting it to inert RhI_3 . It seems unpractical for actual application. Besides, more details need to be provided, such as the recovery percentage and the reactivation pathway of the catalyst.

Response: We thank the reviewer for the insightful comments regarding Rh recovery and catalyst reuse. We fully agree that, given the high cost of Rh, efficient recovery and reusability of the catalyst are critical considerations.

It is important to note that conversion of Rh species to RhI_3 is not required for catalyst reuse under the studied reaction conditions. During catalysis, Rh remains fully dissolved in the homogeneous reaction mixture and retains full catalytic activity. Formation of RhI_3 occurs only after removal of CO and volatile solvent components and therefore does not represent the operative catalytic species. After reaction completion, solvent recovery, and product isolation, the majority of Rh remains in the non-volatile residue, which could, in principle, be directly reused. Although industrial-scale operation was not evaluated, this behavior is conceptually consistent with established principles of Rh retention in liquid-phase carbonylation processes (RU Fuanin et al., JPS581974B).

Laboratory RhI₃-based recycling procedure: After completion of the reaction, the crude mixture was centrifuged to separate the solid fraction (*Solid a*) from the liquid phase (*liquid a*). *Solid a* was washed with aqueous HI to dissolve any trace Rh precipitate that might be present, and the wash solution was collected as *liquid b*. *Liquid a* was distilled to recover the HI/HFIP mixture, which was directly reused in the repeated catalytic cycle. The remaining liquid was evaporated to recover PA. The final residue, containing Rh complexes, was combined with *Liquid b* (denoted as *Liquid c*) and thermally aged to promote precipitation of RhI₃. The mixture was subsequently centrifuged to isolate solid RhI₃. Using this method, more than 95% of the total Rh was recovered, with the remaining loss attributed to unavoidable handling and analytical uncertainties inherent to small-scale experiments. Under standard reaction conditions, the recycled Rh catalyst (Rh-cyclic) exhibited activity and selectivity comparable to fresh RhCl₃ and simulated recycled RhI₃ (prepared by thermal aging of RhCl₃ in excess aqueous HI) when used with fresh HFIP and HI, confirming efficient reactivation and preservation of catalytic integrity (Table R3).

Reactivation pathway: Upon re-exposure to CO and HI under the reaction conditions, isolated RhI₃ is converted in situ into catalytically active Rh–iodide carbonyl species (H_zRh(CO)_xI_y) (RU Fuanin et al., JPS581974B; *Coord. Chem. Rev.*, **2003**, 243, 125–142; *J. Mol. Catal.*, **1987**, 39, 115–136). This conversion occurs via coordination of CO and HI to Rh, which restores the Rh center to its homogeneous, catalytically competent state. The resulting Rh–iodide carbonyl complexes are well-known active species in liquid-phase carbonylation reactions, consistent with established Rh carbonyl chemistry. This reactivation ensures that the recycled Rh catalyst retains comparable activity and selectivity to fresh RhCl₃.

Accordingly, isolation of Rh as RhI₃ was employed solely as a laboratory-scale, quantitative method to recover and account for the total Rh content, representing a conservative, worst-case scenario for catalyst recovery rather than an intrinsic or necessary feature of the catalytic operation. Because this RhI₃-based protocol represents an extreme case in which all Rh is deliberately isolated for laboratory-scale mass-balance analysis rather than retained in the catalytic phase, the corresponding

life-cycle assessment is conservative and likely overestimates Rh-related impacts. Importantly, this conservative treatment does not affect the qualitative conclusions regarding the robustness and recyclability of the catalytic system.

Revision made: Details of the Rh recovery procedure, recovery efficiency, catalyst reactivation pathway, and recycling performance have been added to the **revised Supplementary Information** (Section 5.1 Fig. S10 and Table S4).

Figure R7. Rhodium -catalyzed cyclic processes within the laboratory setting.

Table R3. Reaction performance of different anionic rhodium salts.

No.	Rh	I ⁻	TPA Yield (%)	PA Yield (%)	EG Yield (%)
1	RhCl ₃	CH ₃ I	>99	96.0	0.0
2	RhI ₃	HI	>99	97.6	0.0
3	Rh-cyclic	HI	>99	95.9	0.0

Standard reaction conditions: PET (0.192g, 1 mmol structural unit), HFIP (2 mL), H₂O (0.5 mL), CH₃I (0.5 mmol) or HI (0.4 mmol), CO (2 MPa) and catalyst (RhCl₃, 5 mol%) at 170°C for 12 h.

Comments 2: Second, iodide reagent is also kind of expensive, the author needs to address how to separate and recycle the iodide additive.

Response: We thank the reviewer for raising this important point regarding the separation and recycling of the iodide additive. In this system, iodide is distributed between two fractions after the reaction. The major fraction is present as HI, which is

dissolves in HFIP and remains volatile, while the remainder is retained within the non-volatile Rh-containing catalytic complex. Consequently, iodide recycling is intrinsically integrated with the recycling of HFIP and the Rh catalyst. In line with Comment 1, for laboratory demonstration, the RhI_3 -based recovery procedure was employed to illustrate iodide recycling. The experimental procedure is as follows:

After completion of the reaction, the crude mixture was centrifuged to separate the solid fraction (*Solid a*) from the liquid phase (*liquid a*). *Solid a* was washed with aqueous HI to dissolve any trace Rh precipitate that might be present, and the resulting suspension was centrifuged. The recovered solid was then recrystallized to afford purified TPA, and the wash solution was collected as *Liquid b*. *Liquid a* was distilled to recover the HI/HFIP mixture, which was directly reused in the repeated catalytic cycle. The remaining liquid was evaporated to recover PA. The final residue, containing Rh complexes, was combined with *Liquid b* (denoted as *Liquid c*) and thermally aged to promote precipitation of RhI_3 . The recovered aqueous HI solution could be reused in the subsequent TPA washing step. It is worth noting that this aqueous HI is specifically reserved for the TPA purification cycle and does not enter the catalytic reaction. The resulting non-volatile Rh-containing residue including RhI_3 , together with the recovered HI/HFIP mixture, was directly reused in subsequent repeated catalytic cycle without the addition of extra iodide. Using this approach, the system delivered 98% TPA yield and 90.9% PA yield. Although minor iodide loss is unavoidable under laboratory-scale, non-closed recycling conditions, these results demonstrate that the iodide additive is largely retained and can be efficiently recycled within the present system. This laboratory recovery protocol provides a conservative mass-balance demonstration and aligns with the well-established retention and reactivation behavior of Rh/iodide species in liquid-phase carbonylation systems.

Revision made: The iodide recycling procedure has been added to **the revised Supplementary Information** (Section 5.3, Fig. S10).

Figure R11. The cyclic process of the PET carbonylolytic reaction within the laboratory setting.

Comments 3: Third, the author used hexafluoroisopropanol [HFIP] as solvent for its good properties. However, it is an expensive solvent and have serious environmental impact, so that its recovery needs to be addressed.

Response: We thank the reviewer for highlighting the cost and environmental considerations associated with HFIP. We agree that demonstrating efficient solvent recovery and reuse is essential for assessing the practical viability of this process. Accordingly, we conducted solvent-recycling experiments to evaluate the recovery efficiency and chemical stability of HFIP under the reaction conditions.

HFIP recycling procedure: After completion of the reaction, the crude mixture was centrifuged to separate the solid fraction from the liquid phase. The liquid phase was then subjected to low-temperature distillation to recover an HI/HFIP mixed solution, which was directly reused in subsequent catalytic cycle. Because HFIP and HI function cooperatively as the reaction medium, separation of HI from HFIP is neither required nor desirable for effective solvent circulation. Moreover, the co-recycling of solvent and iodide promoter is consistent with established industrial practice in Monsanto/Cativa-type carbonylation processes.

HFIP was recovered after each run with near-quantitative efficiency (>99%), and ^1H , ^{13}C and ^{19}F NMR analyses (see also our response to Reviewer 2, Comment 5) confirmed that the recovered solvent retained high chemical purity without detectable degradation. Reuse of the recycled HFIP with fresh RhCl_3 resulted in no observable

loss of catalytic activity or product selectivity, demonstrating that HFIP can be efficiently recycled and reused without compromising performance (Table R4). These results establish the practical viability of HFIP recycling, and exploration of alternative greener solvent systems remains an important direction for future work.

Revision made: In the revised Supplementary Information, the NMR spectra of fresh HFIP and recycled HFIP are provided in Figures S11–S13, and the corresponding catalytic performance data are summarized in Table S5.

Table R4. Recyclability Test of HFIP

Cycle Number	Temp.	Sol.	TPA Yield (%)	PA Yield (%)
1	170°C	HFIP	99	96.3
2	170°C	HFIP	99	96.7
3	170°C	HFIP	99	96.4

Standard reaction conditions: PET (0.192g, 1 mmol structural unit), HFIP (2 mL), H₂O (0.5 mL), CH₃I (0.5 mmol), CO (2 MPa) and catalyst (RhCl₃, 5 mol%) at 170°C for 12 h.

Comments 4: Fourth, the author claims that the carbonylation mechanism is through the ethylene insertion into [Rh(CO)₂I₂]⁻ active catalytic species. Actually, the well-recognized active catalytic species is supposed to be [HRh(CO)I₃]⁻. You can follow a recently published paper *J. Am. Chem. Soc.* **2025**, 147, 38, 34589–34600.

Response: We thank the reviewer for the important mechanistic clarification. We fully agree that, according to well-established carbonylation chemistry, the hydride complex [HRh(CO)I₃]⁻ is the widely recognized active catalytic species, as comprehensively documented in the recent report by *J. Am. Chem. Soc.* **2025**, 147, 34589–34600 and related literature (M Beller, *Catalytic carbonylation reactions*, **2006**, 18, Springer Science & Business Media; *J. Mol. Catal. A: Chem.*, **1995**, 104, 17–85; *J. Catal.*, **2014**, 319, 211–219; *ACS Catal.*, **2022**, 4203–4215).

In light of this, we have revised the mechanistic description throughout the

manuscript to identify $[\text{HRh}(\text{CO})\text{I}_3]^-$ as the primary active species governing ethylene insertion and subsequent carbonylation steps. Consistent with this revision, we have reconducted the DFT calculations using $[\text{HRh}(\text{CO})\text{I}_3]^-$ as the catalytic center, and the updated results are now presented in the revised manuscript and Supplementary Information. The new DFT results confirm that ethylene coordination and insertion into the Rh–H bond of $[\text{HRh}(\text{CO})\text{I}_3]^-$ proceed with accessible activation barriers under the reaction conditions, followed by CO migration and hydrolysis to afford the observed products, in full agreement with the established hydride-based catalytic cycle.

Revisions made: The mechanistic section and catalytic cycle have been updated to reflect the hydride-mediated pathway via $[\text{HRh}(\text{CO})\text{I}_3]^-$, and the DFT analysis has been revised accordingly (Figure 4E in the revised manuscript and Section 4.1 in the revised Supplementary Information).

Figure R12. Revised Figure 4E of the manuscript. Gibbs free-energy profile for the carbonylation of 1,2-diiodoethane.

Reviewer #1 (Remarks to the Author):

After reviewing the revisions made by Liu et al., I am fully supportive of the publication of the manuscript in Nature Communications. The authors have fully addressed the three reviewers' concerns by not only adding additional clarifications where needed, but by also providing additional experimental data which support their claims made throughout the manuscript.

Response: We sincerely thank you for your careful review and thoughtful evaluation. Your insightful comments and suggestions have been invaluable in improving both the clarity and quality of our work. We are truly honored by your positive recommendation and greatly appreciate your recognition and support.

Reviewer #2 (Remarks to the Author):

This manuscript can be accepted as it is.

Response: Thank you very much for your careful review and encouraging assessment of our work. Your feedback has significantly contributed to improving the clarity and presentation of our manuscript. We are deeply honored by your positive evaluation and sincerely appreciate your constructive guidance.

Reviewer #3 (Remarks to the Author):

The author has answered most of the questions. But I think the answers to the reviewer's comments are not sufficient, the following issues are still need to be addressed.

Response: We sincerely thank you for the insightful and constructive feedback. We greatly appreciate your recognition of our efforts to address the initial concerns and your emphasis on the remaining key aspects. We fully acknowledge that these are critical factors that require careful consideration, and each has been thoroughly addressed in the revised manuscript. Below, we provide detailed responses to the specific points you raised.

Comments 1: The author claimed in the response that “A sharp increase in activity was observed upon increasing CH₃I loading from sub-threshold to iodide-rich conditions, followed by a plateau upon moderate further increases, indicating that a minimum iodide level is required to establish the active Rh catalytic cycle rather than a linear dependence on CH₃I loading.” However, only three data provided in not enough to draw this conclusion. I think more data should be added to draw a relationship line. Besides, the author should use this data to correlate to the catalytically active Rh – iodide species and give a reasonable explanation.

Response: We thank the reviewer for this insightful comment and fully agree that the initial presentation of the CH₃I dependence, based on a limited number of data points within the critical concentration window, was insufficient to rigorously substantiate the proposed non-linear relationship. In response to the reviewer’s suggestion, we have now inserted additional data points specifically within the key concentration region (0.10–0.30 mmol per 1 mmol PET), where the sharp change in catalytic behavior occurs, thereby enabling a clearer construction of the activity–iodide relationship (Table R1).

The refined dataset within this critical window reveals a pronounced non-linear dependence of catalytic performance on CH₃I loading. At low CH₃I levels (<0.10 mmol per 1 mmol PET), PA formation is strongly suppressed and EG remains the dominant product. Upon increasing CH₃I into an intermediate regime, PA yield increases sharply over a narrow concentration range (0.10–0.20 mmol per 1 mmol PET). Further increases in CH₃I result in only marginal changes in activity and selectivity, indicative of saturation-type behavior (0.20–0.50 mmol per 1 mmol PET), while at higher CH₃I loadings (~0.80 mmol per 1 mmol PET) a decrease in PA yield is observed.

This non-linear trend can be directly correlated with changes in steady-state Rh–iodide speciation arising from competitive coordination between iodide and CO at the Rh center. At sub-critical iodide activities, iodide is unable to effectively compete with CO for coordination, and Rh is predominantly present in iodide-deficient or neutral

carbonyl environments that are ineffective for oxidative addition and subsequent CO insertion, resulting in low PA formation. Increasing CH₃I raises iodide activity in solution, enabling iodide to successfully compete with CO and stabilize mixed anionic Rh–iodide–carbonyl species (commonly invoked in Rh-catalyzed carbonylation cycles, such as [RhI₂(CO)₂]⁻ or [HRh(CO)I₃]⁻-type intermediates), which are catalytically competent for oxidative addition and CO insertion.

Once these anionic Rh–iodide–carbonyl species become the dominant steady-state form, the carbonylation cycle is fully established and PA formation proceeds efficiently. Within this CH₃I concentration window, catalytic performance remains essentially constant, giving rise to the observed activity plateau. At higher CH₃I loadings, however, excess iodide overcompetes with CO for coordination and drives Rh speciation toward iodide-saturated, CO-deficient environments, limiting the availability of Rh–CO species required for CO insertion. As a result, further increases in CH₃I do not enhance and may even suppress catalytic activity. Thus, the observed CH₃I dependence reflects modulation of the steady-state population of catalytically active Rh–iodide species through iodide–CO coordination balance, rather than a direct kinetic dependence on CH₃I concentration itself. While this interpretation is based on catalytic trends rather than direct in situ speciation measurements, it is fully consistent with established Rh–iodide carbonylation chemistry. (*Catal. Sci. Technol.*, 2022, 12, 664-673; *ACS Catal.*, 2021, 11, 7249-7256; *J. Catal.*, 2014, 319, 211-219; *Sci. China, Ser. B: Chem*, 2004, 47, 41-49; *J. Catal.*, 2015, 325, 1-8; *The Canadian Journal of Chemical Engineering*, 2021, 99, 1137-1145.)

Revisions made: In the **revised manuscript**, the discussion has been rephrased to explicitly correlate the CH₃I dependence with the formation of catalytically active Rh–iodide species. Specifically, we now state that “*As CH₃I is increased into an intermediate regime, the steady-state population of catalytically competent anionic Rh–iodide–carbonyl species is established, leading to a sharp rise in PA formation (Table S2). Further increases in CH₃I result in a plateau in activity and selectivity, indicative of saturation of the active Rh species. At much higher CH₃I loadings, excess iodide overcompetes with CO for coordination, driving Rh speciation toward iodide-*

saturated CO-deficient environments and limiting CO insertion, which reduces PA yield (Fig. 2A). This behavior is fully consistent with established Rh-iodide carbonylation chemistry”

In addition, Table S2 in the **revised Supplementary Information Section 2.6** includes the newly added data points within the critical CH₃I concentration window, providing explicit experimental support for the non-linear dependence and the requirement of a minimum iodide level to establish the active Rh catalytic cycle.

Table R1. Reaction performance of different CH₃I content.

No.	CH ₃ I loading	TPA Yield (%)	PA Yield (%)
1	0.10	95.1	0.33
2	0.12	96.2	5.0
3	0.15	96.8	6.8
4	0.18	97.3	17.3
5	0.20	99.0	83.0
6	0.30	97.5	85.4

Standard reaction conditions: PET (0.192g, 1 mmol structural unit), solvent (2 mL), H₂O (0.5 mL), CH₃I (0.5 mmol), CO (2 MPa) and catalyst (RhCl₃, 5 mol%) at 170°C for 12 h.

Comments 2: The author mentioned in the response that “Notably, all PET samples were manually cut into small pieces without further powdering or micronization, thereby minimizing potential structural modification associated with intensive mechanical pretreatments such as ball milling.” Is there any study on the effect of the PET sample size? It seems unreasonable that the result is irrelevant to the substrate size.

Response: We thank the reviewer for this insightful comment and fully agree that, in general, the particle size of polymeric substrates can significantly influence reaction kinetics by affecting surface area and mass-transfer processes. To directly address this concern, we have performed additional experiments using PET bottle flakes with different dimensions (0.3×0.3 , 0.5×0.5 , 1.0×1.0 , 1.5×1.5 , 2.0×2.0 cm² and randomly irregular mixture) under otherwise identical carbonylation conditions (Fig. R1 and Fig. R2). The results show that the yields of terephthalic acid (TPA) and propionic acid (PA) remain essentially unchanged across this particle-size range, indicating that PET particle size exhibits negligible dependence within the reaction regime investigated in this work.

This behavior can be rationalized by the nature of the reaction medium and mechanism. HFIP is a strong solvent for PET and can readily swell and dissolve the polymer under the reaction conditions. This rapid solvent penetration substantially alleviates mass-transfer limitations associated with solid–liquid interfaces, such that the transformation is not strictly governed by heterogeneous interfacial reactions, particularly after the initial solvation and depolymerization stages. Once PET is solubilized and depolymerized to molecular intermediates (e.g., EG), the subsequent carbonylation step proceeds predominantly via homogeneous molecular pathways. Consequently, the overall reaction rate and selectivity are governed primarily by the homogeneous carbonylation chemistry rather than by the specific surface area of the solid polymer.

It is noteworthy that this observation is specific to the present HFIP-mediated system and does not imply that PET particle size is universally unimportant across all depolymerization processes. Under the conditions employed here, efficient conversion

can be achieved without energy-intensive mechanical pretreatments such as ball milling or micronization.

Revisions made: Additional particle-size-dependent experiments have been included in the revised **Supplementary Information Section 2.8** (Fig. S7 and Table S3). The revised manuscript text has been revised to clarify that PET powdering is unnecessary for efficient conversion specifically under the HFIP-mediated reaction conditions studied here, due to the strong solubility of HFIP toward PET, thereby avoiding any implication of universal size-independence (p. 4, line 16).

Fig. R1 Different sizes of PET bottle flakes. (A-B) $0.3 \times 0.3 \text{ cm}^2$; (C-D) $0.5 \times 0.5 \text{ cm}^2$; (E-F) $1.0 \times 1.0 \text{ cm}^2$; (G-H) $1.5 \times 1.5 \text{ cm}^2$; (I-J) $2.0 \times 2.0 \text{ cm}^2$ (These data have been added to Fig. S7).

Table R2. Reaction performance of different PET samples sizes. (These data have been added to Table. S3)

No.	PET samples size	TPA Yield (%)	PA Yield (%)
1	2.0 × 2.0	94.6	93.6
2	1.5 × 1.5	91.2	90.6
3	1.0 × 1.0	95.1	92.5
4	0.5 × 0.5	92.7	91.6
5	0.3 × 0.3	96.0	92.2
6	randomly irregular mixture	94.1	92.1

Standard reaction conditions: PET (0.192g, 1 mmol structural unit), solvent (2 mL), H₂O (0.5 mL), CH₃I (0.5 mmol), CO (2 MPa) and catalyst (RhCl₃, 5 mol%) at 170°C for 12 h.

Comments 3: The author described a complex procedure for the catalyst recycling.

First, only 95% of Rh can be recycled, which would lead to significant high cost. A reasonable production cost needs to be provided. Second, the recycling efficiency of iodide species should be calculated, which also contribute to the high cost of the procedure.

Response: We thank the reviewer for raising these important concerns, and we fully agree that efficient resource retention and economic viability are critical for practical applications.

Rh recovery and cost implications: The recovery of “~95% Rh as RhI_3 ” should not be directly equated with an intrinsic ~5% catalyst loss, as it does not represent the exact behavior of the catalytic system under reaction conditions. After TPA separation, distillation of volatile components (HFIP/HI and PA) of the reaction mixture leave a non-volatile Rh-containing residue that constitutes the actual recycled catalytic phase. ICP–MS analysis of this residue indicates near-quantitative Rh retention (~99%, within analytical error), and direct reuse of this Rh-containing residue in subsequent cycles, without deliberate precipitation of RhI_3 , delivers 97% PA yield, consistent with fresh runs, demonstrating effective recyclability and catalytic stability under closed-system operation. In a realistic process design, Rh would therefore be continuously retained in this non-volatile catalytic residue and recycled following distillation of volatile solvents and products, rather than being isolated or converted to RhI_3 between cycles. This behavior aligns with established industrial practice in liquid-phase Rh-catalyzed carbonylation, where Rh is maintained in a high-boiling catalytic phase and recovered primarily via product distillation. The ~95% Rh recovery was determined gravimetrically after intentional RhI_3 isolation, which was employed solely as a laboratory-scale method for rigorous mass-balance accounting under non-closed handling conditions. This procedure involves introducing additional transfer, washing, and phase-separation steps that are not required in a practical process and introduces experimental errors at small scale. Importantly, isolation of Rh as RhI_3 is an extreme laboratory representation of the complex mixture of Rh species present in the catalytic residue, rather than the actual form in a practical process.

Based on the near-quantitative Rh retention measured by ICP–MS and the successful direct reuse of the Rh-containing residue, the effective Rh cost contribution is expected to be only a small fraction of overall process costs in a closed-system

operation. The primary determinant of catalyst-related cost is therefore not intrinsic chemical consumption, but the process to retain Rh in a non-volatile catalytic residue and recycle it directly after product distillation, as confirmed experimentally. For TEA purposes, RhI_3 is adopted as a representative Rh species to simplify accounting and modeling, while intrinsic Rh loss is neglected in accordance with the ICP–MS results. This provides a conservative, order-of-magnitude estimate rather than a fully detailed process model. Consistent with extensive literature, Rh–iodide carbonylation catalysts sustain high-turnover operation through continuous retention, such that the Rh loading should be regarded as a retained catalytic inventory rather than a consumable input (*Coord. Chem. Rev.*, 2003, 243, 125–142; *J. Mol. Catal.*, 1987, 39, 115–136; *Adv. Catal.*, 2010, 53, 1–45). In parallel, we are pursuing further cost optimization via two complementary strategies: development of readily separable heterogeneous catalytic systems, and exploration of low-cost, non-precious metal alternatives to Rh.

Iodide recycling efficiency: In this system, iodide recovery is intrinsically coupled with the recovery of both the solvent and the catalyst. After reaction, most of the iodide remains dissolved in HFIP as HI and can be recovered by distillation and directly reused, while the remaining iodide is retained in the non-volatile Rh-containing catalytic residue. In the laboratory RhI_3 -based recovery protocol, the iodide associated with Rh is quantitatively preserved and returned to the system upon catalyst reactivation. Iodometric titration of the distilled HI/HFIP solution (**revised Supporting Information** Section 4.4), together with the independently determined 99% Rh retention, indicates an overall iodide recovery of >93% (defined as total iodide recovered relative to iodide initially charged) under non-closed laboratory operating conditions. The small losses observed are primarily due to open distillation and manual handling rather than intrinsic chemistry and represent conservative laboratory artifacts. Multiple catalytic cycles were carried out without adding any external iodide, demonstrating excellent practical recyclability. Because iodide functions catalytically and is effectively recovered with both the solvent and the catalyst, these minor lab-scale losses have a negligible impact on total operating costs. In industrial-scale closed or semi-closed operations, such handling-related losses can be further minimized, further reducing the effective cost contribution of iodide.

About the cost: From a practical perspective, the present recycling experiments are intended to demonstrate catalyst and promoter retention, reactivation capability, and mechanistic robustness, rather than to provide a full industrial process design. In

our system, Rh is recovered in a non-volatile catalytic residue, while iodide is largely retained in the volatile HFIP/HI phase with well-established methods available to minimize volatile losses. Based on our experiments and established industrial practice in liquid-phase Rh carbonylation, material losses of both Rh and iodide are expected to be minimal under closed or semi-closed operation. Accordingly, Rh can be reasonably regarded as a retained catalytic inventory rather than a consumable reagent. Consequently, the dominant economic factors are expected to be process operation and energy consumption, which have been considered qualitatively in our current techno-economic assessment using RhI_3 as a conservative reference. Exact quantitative cost estimation under industrial conditions would require scaled-up experiments and detailed engineering studies, which are beyond the scope of the current work. Future work will focus on process intensification, simplified catalyst recovery strategies, and alternative catalyst designs to further enhance operational efficiency and economic competitiveness.

Revisions made: The discussion section in the **revised manuscript** now includes a sentence highlighting limitations and future directions: *“large-scale deployment requires advances in cost-efficient non-noble catalysts and robust performance with highly mixed or contaminated feedstocks, with catalyst recovery and operational efficiency being particularly critical.”* **Revised Supporting Information** Sections 4.1 and 4.3 report ICP–MS–determined Rh retention (~99%, within analytical error) and iodide recovery (>93%), and Section 4.4 details the iodide quantification method. The following key points are also clarified in the Supporting Information: in practical recycling, Rh is retained in the non-volatile catalytic residue and directly reused, enabling near-quantitative catalyst retention. Under closed or semi-closed operation, where Rh is continuously retained, the effective Rh cost is expected to be dominated by catalyst inventory amortization rather than consumption. RhI_3 is used as a representative Rh species for accounting and modeling purposes, and any minor laboratory-scale loss does not impact overall production costs, as Rh retention is effectively quantitative.